# Cell-cycle dependent DNA repair and replication unifies patterns of chromosome instability

Bingxin Lu ●[1,2,3,4] ✉, Samuel Winnall ●[1], William Cross[1,5] & Chris P. Barnes ●[1,2] ✉

Chromosomal instability (CIN) is pervasive in human tumours and often leads to structural or numerical chromosomal aberrations. Somatic structural variants (SVs) are intimately related to copy number alterations but the two types of variant are often studied independently. Additionally, despite numerous studies on detecting various SV patterns, there are still no general quantitative models of SV generation. To address this issue, we develop a computational cell-cycle model for the generation of SVs from end-joining repair and replication after double-strand break formation. Our model provides quantitative information on the relationship between breakage fusion bridge cycle, chromothripsis, seismic amplification, and extra-chromosomal circular DNA. Given whole-genome sequencing data, the model also allows us to infer important parameters in SV generation with Bayesian inference. Our quantitative framework unifies disparate genomic patterns resulted from CIN, provides a null mutational model for SV, and reveals deeper insights into the impact of genome rearrangement on tumour evolution.

Chromosomal instability (CIN) is a major form of genome instability widely present in human tumours, which often leads to structural or numerical chromosomal aberrations and plays an important role in cancer evolution[1,2]. Somatic structural variants (SVs) are large genomic rearrangements showing a variety of patterns and have been classified into different types, such as duplication, deletion, inversion, translocation, and other complex variants[3–10]. Duplication and deletion are also called copy number alterations (CNAs), which include whole genome doubling (WGD) as well and are often studied separately from SVs[7,9,11,12]. Typical complex SVs include breakage-fusion-bridge (BFB) cycle which is an ongoing process of chromosome bridge breakage[13], chromothripsis in which genomic regions on one or a few chromosomes are shattered and restitched with segment loss[14], extrachromosomal circular DNA (ecDNA)[15], seismic amplification which are high-level amplifications (at least 5 copies for diploid and 9 copies for polyploid genomes) with at least 14 internal SVs[8], and chromoplexy which is a closed chain of inter-chromosomal translocations involving multiple chromosomes[16]. From these various patterns, it is important to investigate the underlying mechanisms, which will provide further insights into CIN and cancer evolution and potentially inform treatment[5,17–21].

Each SV manifests as the spatial apposition of breakpoints[17] that is caused by the interplay of DNA damage, replication errors, and repair pathways occurring throughout the cell cycle[22]. Many DNA damage events involve double-strand breaks (DSBs), which can be repaired by two broad mechanisms: homologous recombination (HR), which has high fidelity but requires extensive sequence homology at the breakpoints; or non-homologous end joining (NHEJ) that needs no sequence homology but is error-prone[6,23,24]. Indeed, NHEJ is known to play a major role in the formation of focal deletions and translocations[25], chromothripsis[26–30], ecDNAs[15], and various simple and complex SVs after chromosome segregation errors[31]. How erroneous DNA repair mechanisms directly contribute to the complex SVs observed in cancer genomes still remains unclear[21,24].

[1]Department of Cell and Developmental Biology, University College London, Gower Street, London, UK. [2]UCL Genetics Institute, University College London, Gower Street, London, UK. [3]School of Biosciences, University of Surrey, Stag Hill, Guildford, UK. [4]Surrey Institute for People-Centred Artificial Intelligence, University of Surrey, Stag Hill, Guildford, UK. [5]Rare Malignancies and Cancer Evolution Laboratory, School of Biological Sciences, University of Reading, Whiteknights, Reading, UK. ✉e-mail: b.lu@surrey.ac.uk; christopher.barnes@ucl.ac.uk

This question cannot be answered by experimental approaches alone due to technical limitations[20,32,33] and requires quantitative models that can encode our knowledge of the underlying biological processes. The quantitative models enable long-term simulation to study evolutionary processes inaccessible by experiments and provide a basis for statistical inference. A few models have been developed to investigate CNAs[2,33–36] and certain types of SVs, such as chromothripsis[14,37], chromoplexy[16], ecDNA[15,16], neochromosome[38], and seismic amplification[8]. However, these models focused on specific SV types and local genomic regions, therefore missing the underlying processes that relate all of these variant classes. This is like studying wind speed, precipitation, thunderstorms, and tornadoes without understanding changes in air temperature and air pressure.

To address this gap, we developed a conceptually simple, yet unifying framework to model the dynamics of SV formation. Our multi-scale computational model consists of simulated cells progressing through cell cycles via a stochastic birth-death branching process where fitness effects can be captured. A cell undergoes random DSB repair through end-joining and genome replication in interphase and then correctly or incorrectly divides in mitosis. Using our model, we provide direct quantitative evidence that different types of SVs are generated during the same ongoing DNA repair and replication process, including BFB cycle, chromothripsis, ecDNA, and seismic amplification. We then apply the model with Bayesian inference techniques to infer important parameters in single-cell and bulk whole-genome sequencing data, such as DSB rate per cycle, probability of WGD per cell, and otherwise inaccessible information including mean number of chromosome fusions per cycle and mean number of ecDNAs per cell.

## Results

### A stochastic model of SV generation across cell cycles

To model SV generation within a cell, we introduced DSBs and repaired them with random end-joining across the cell cycle. Cells divided according to a stochastic birth-death branching process (Methods, Fig. 1a, Supplementary Table 1). For simplicity, we did not explicitly consider sequence homology at breakpoints when repairing DSBs, and hence the erroneous repair in our model is considered as mainly deriving from NHEJ, which was previously shown to generate many SVs in cancer genomes[6]. Some SVs such as tandem duplications may be generated by homologous repair deficiency (HRD) or DNA replication errors[25,39]. However, since only around 12% of cancers show tandem duplication phenotype[40], we did not include these processes in our model.

We also assumed that a fraction of DSBs may remain unrepaired due to checkpoint deregulation such as TP53 mutation[22]. As SVs may bring selection advantages or disadvantages to cancer cells, we incorporated a simple model of selection based on the density of oncogenes (OGs) and tumour suppressor genes (TSGs)[2,41], in which karyotypes with a higher oncogenic propensity were favoured.

To keep track of breakpoint joining on the complete genome that becomes more shattered with additional DSBs across cell cycles, we used a diploid interval adjacency graph $G = (V, E)$ to represent each genome[42], where $V$ represents the set of breakpoints (nodes) and $E$ represents their adjacencies (edges) (Fig. 1b). There are three types of adjacencies according to the positions in the reference genome, including an interval adjacency between two endpoints of a genomic interval, a reference adjacency connecting two adjacent intervals, and a variant adjacency caused by SVs which connects two non-adjacent intervals. By using a graph representation to capture the complexity of SVs in a complete genome, we can generate realistic data including CNAs and SVs for all the cells in the final population.

To illustrate our model in simulating SV formation over time, we show the accumulation of SVs in two cell cycles under neutral evolution (Methods, Fig. 1c). The formation of SVs is clearly shown in a step-

wise way at the haplotype level, helping to understand the link between DNA repair and replication with different types of SVs.

### The simple repair process model explains the formation of complex SVs

With appropriate parameters, our model enables forward simulations which generate realistic inter-chromosomal and intra-chromosomal rearrangements, such as BFB cycle, chromothripsis, focal and seismic amplification, ecDNA, chromoplexy, and parallel haplotype-specific CNA[20,21,43] (Methods, Fig. 2a, Supplementary Table 1).

In our model, the fusion of sister chromatids of a telomere-lacking chromosome in interphase leads to the forming of a chromosome bridge and the asymmetric bridge breakage in mitosis (Fig. 1a), which was shown to generate either simple breaks or local fragmentation that trigger iterative cycles of complex mutational events including chromothripsis interwoven with BFB cycles[20]. To validate this phenomenon, we introduced one unrepaired DSB on a random chromosome followed by either simple breaks or local fragmentation in the simulations. The simulation with simple breaks recaptured the typical features of a BFB cycle, including staircase-like focal inverted duplications or fold back inversions (FBIs) with clustered breakpoints adjacent to terminal losses or gains on the same homologue[21,43] (Fig. 2b, c). Under selection, cells with higher-level amplifications seem slightly favoured, which is likely due to a few cells with higher copy numbers of OGs such as CARD11 and EGFR (Supplementary Table 2). The results also directly demonstrate that an ongoing BFB cycle initiated by the formation of a chromosome bridge caused progressive shorting of a homologue and serrate structural variation (SSV) (Fig. 2b). These two patterns were recently reported in bulk or single-cell whole-genome sequencing data and suggested to reflect ongoing BFB cycles or mutational process[20,43]. With local fragmentation, the random segregation of highly broken dicentric chromosomes led to a number of chromothripsis, ecDNAs, and seismic amplifications which started appearing at similar times with or without WGD (Figs. 2d,e, Supplementary Fig. 1–3, Supplementary Table 3). Furthermore, cells with both chromothripsis and ecDNAs had slightly higher numbers of ecDNAs, with a significant increase of ecDNAs and chromothripsis following local fragmentation (Supplementary Fig. 4). These results suggest that the emergence of ecDNA is influenced by local fragmentation, which was shown to potentially trigger chromothripsis[20]. Repeated simulations incorporating WGD also indicated a significantly higher occurrence of ecDNAs and chromothripsis per cell in WGD-positive samples (Supplementary Fig. 5–6). The sizes of simulated ecDNAs mainly ranged from hundreds of bases to several megabases, consistent with those reported in literature[44], and there were slightly more different types of ecDNAs per cell under positive selection (Supplementary Fig. 7). Due to weak selection forces, the numbers of complex SVs generated under neutral evolution and selection were similar (Supplementary Table 4). Our model provides the earliest in silico evidence that BFB cycles, chromothripsis, and ecDNAs are closely related by erroneous DNA repair (likely from NHEJ) and replication at the whole genome level. This is consistent with the known experimental discovery that BFB cycles and chromothripsis trigger ecDNA formation and amplification along with NHEJ inhibitors[15]. Our model also suggests that chromothripsis can contribute to seismic amplification. This agrees with previous work showing that seismic amplification can arise through chromothripsis followed by circular BFB or recombination of ecDNAs[8].

Balanced rearrangements such as chromoplexy were hypothesised to arise from incorrect rejoining of several simultaneous DSBs in multiple chromosomes, which may span several successive events[6,16,19]. To validate this hypothesis, we introduced 10 misrepaired DSBs on random chromosomes during each cell cycle until reaching 5 or 10 cells under neutral evolution (with 10 additional repeats). The results show that chromoplexy appeared within just three or four cell cycles (Fig. 2f, Supplementary Fig. 8–9). Further repeated simulations with

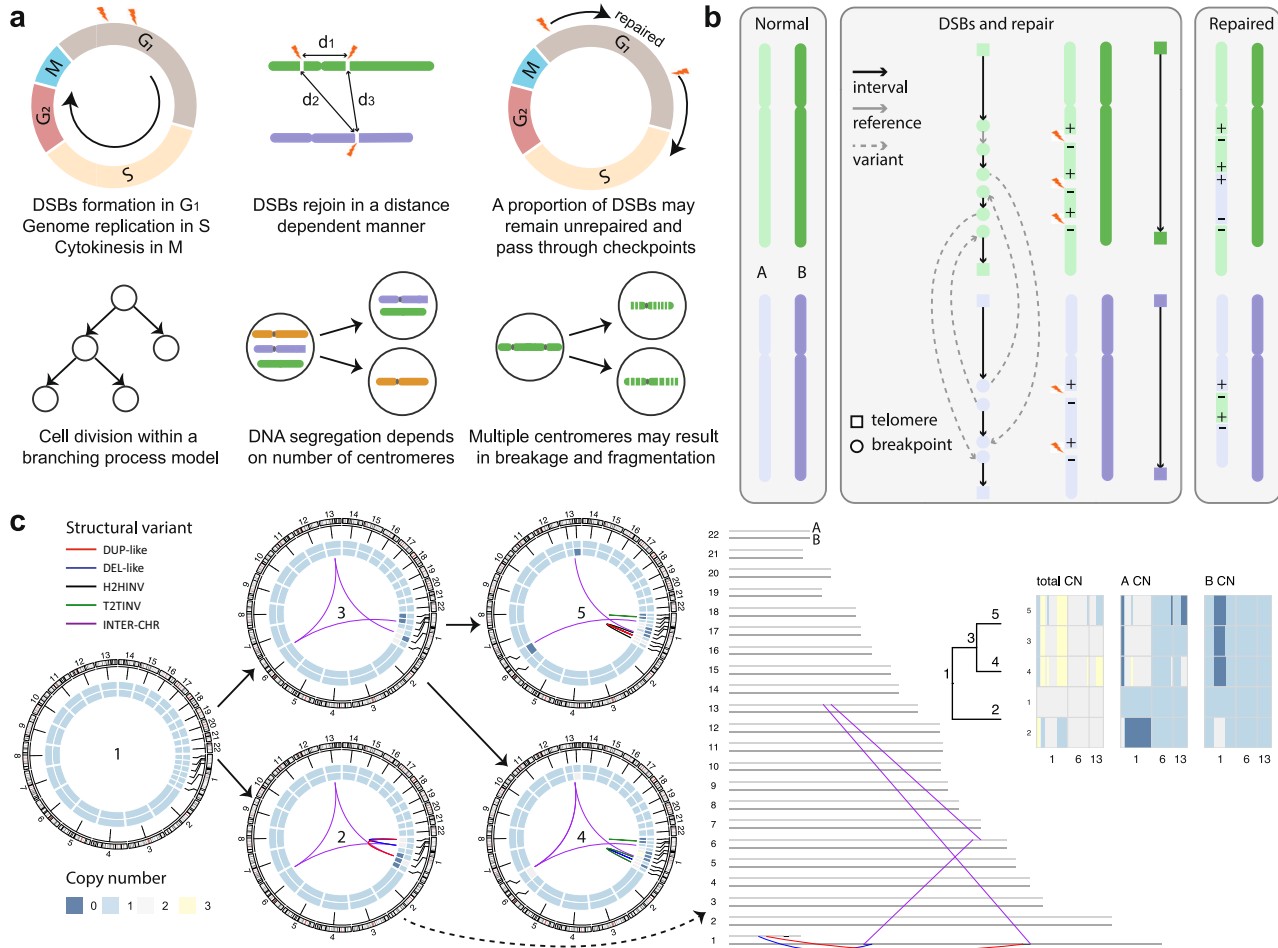

**Fig. 1 | Stochastic cell-cycle model of structural variant (SV) generation from DNA repair and replication. a** Illustration of the cell-cycle model along a branching process. The cell cycle contains interphase and mitosis (*M* phase), where interphase consists of DNA replication with three phases: $G_1$ (growth) in which the cell grows, *S* (synthesis) in which the cell replicates DNA, and $G_2$ (growth) in which the cell grows to prepare for mitosis. In $G_1$, the repairing of the broken chromosomes depends on their linear genomic distance and may form chromosome fusion. Some unrepaired double-strand breaks (DSBs) may lead to sister chromatid fusion after DNA replication in *S* and $G_2$. In *M*, the chromosomes with one centromere are distributed to each daughter cell and the chromosome without centromere is randomly distributed to a daughter cell. The chromosomes with more than one centromere are randomly broken and distributed to the daughter cells, imitating breakage-fusion-bridge (BFB) which may cause local fragmentation. **b** The graph representation of a diploid genome. Left, two pairs of normal chromosomes with light colour indicating homologue A and dark colour indicating homologue B. Middle, one homologue of each chromosome is broken at some breakpoints and rejoined randomly. The normal chromosome is represented by two telomeres connected with an interval edge. The broken and repaired chromosome is represented by the joining of both telomeres and breakpoints with interval, reference, and variant edges. Right, the repaired chromosomes with SVs (translocation and inversion). **c** The accumulation of SVs in two cell cycles starting from a normal cell. Left, the circos plots of all the cells with cell lineage information, where the inner and outer heatmaps within the circle represent copy numbers of homologue A and B respectively. Middle, the haplotype graph of cell 2, where the breakpoints are located at different homologues which are not distinguished in the circos plots. Right, the cell lineage tree and copy numbers of chromosomes with copy number alterations (CNAs). The copy numbers in the two daughter cells after a cell cycle are always reciprocal. DUP-like: tandem duplication-like patterns, DEL-like: deletion-like patterns, H2HINV: head-to-head inversion, T2TINV: tail-to-tail inversion, INTER-CHR: inter-chromosomal SV. Source data are provided as a Source Data file.

fewer misrepaired DSBs per cycle demonstrated that chromoplexy started to appear with more than one DSB per cycle (Methods, Supplementary Fig. 9). The underlying processes of chromoplexy are still largely unknown[21], and experimental studies have recently been done to investigate whether co-localisation of multiple DSBs can stimulate chained inter-chromosomal and intra-chromosomal translocations typical for chromoplexy[45]. Our simulations may help to further understand the processes generating chromoplexy by fitting experimental data in the future. As a DSB occurred on one homologue of the affected chromosome, we also observed parallel evolution of haplotype-specific CNAs that affect different alleles of the same sites in different cells when we continued the simulation until 200 cells (Fig. 2g). Parallel haplotype-specific CNAs have been commonly detected in cancer genomes, which may influence transcription and

make it challenging to compute variant allele frequency in bulk sequencing data[2,43]. Our model provides a straightforward way to simulate them and may further contribute to understanding their roles in cancer evolution.

## Role of cell cycle in formation of complex SVs including chromothripsis
We explored how DSB rate per cycle, percentage of unrepaired DSBs per cycle, and scale of local fragmentation (measured by mean number of DSBs per fragmentation) affected the formation of chromothripsis in one or two cell cycles, which was linked with poor survival or prognosis in cancer patients[46] (Methods, Fig. 3a, Supplementary Table 1). The results suggest that chromothripsis events were generated in one or two cell cycles as a result of chromosome fusions or BFB

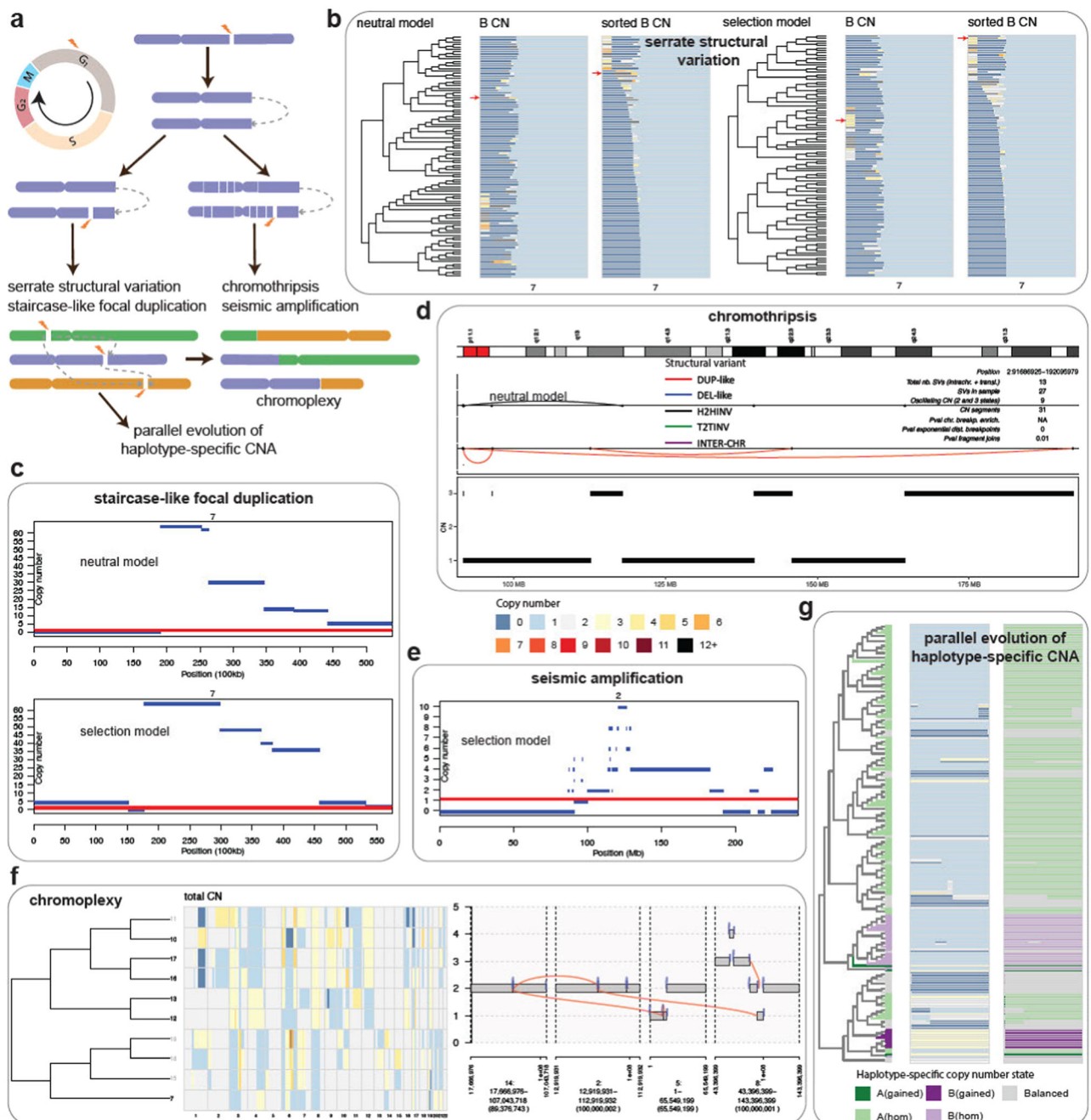

**Fig. 2 | Demonstration of structural variant (SV) patterns generated by the cell-cycle model. a** Overview of the model settings to generate SVs with either one unrepaired double-strand breaks (DSBs) or multiple misrepaired misrepaired DSBs. **b** The cell lineage tree of 100 cells and their corresponding copy numbers on chr7 from the simulation with simple breaks under neutral evolution and selection, where the heatmap of homologue B on the right is sorted to show serrate structural variations (SSVs). **c** The haplotype-specific copy number alterations (CNAs) of two cells (indicated by red arrows in (**b**)) suggest staircase-like focal inverted duplications. **d** Chromothripsis detected in one cell after the eighth cell cycle under neutral

evolution (indicated by the red arrow in Supplementary Fig. 1a). **e** Haplotype-specific CNAs of one cell at the fifth cell cycle with seismic amplification under selection (indicated by the red arrow in Supplementary Fig. 1b). **f** The cell lineage tree of 10 cells and their corresponding copy numbers across all chromosomes from the simulation with multiple misrepaired DSBs, as well as a detected chromoplexy in cell 10. The cells without chromoplexy are shown in grey. **g** The cell lineage tree of 183 cells (excluding 17 cells lacking the pattern) and their corresponding total copy numbers and haplotype-specific copy number states at region (19,444,803, 27,298,459) of chr22. Source data are provided as a Source Data file.

cycles, when multiple DSBs co-located on chromosomes with all of them being misrepaired or a fraction remaining unrepaired (Fig. 3b, c). Even with a larger number of DSBs in one cell cycle, fewer chromothripsis events occurred (Fig. 3c), along with a smaller number of chromosome fusions (Fig. 3b) and ecDNAs (Fig. 3d) and no seismic amplification (Fig. 3e, Supplementary Fig. 10), which all increased in two cell cycles. Therefore, chromothripsis seems more likely to result

from a cascade of mutational events as demonstrated in recent experiments where chromothripsis occurred at a low frequency during the first interphase and became more prevalent during the second cell cycle[20]. The numbers of BFB cycles and ecDNAs generally increased with the number of cell cycles, the DSB rate, the scale of local fragmentation, and the percentage of unrepaired DSBs. The numbers of chromothripsis and seismic amplifications fluctuated sometimes,

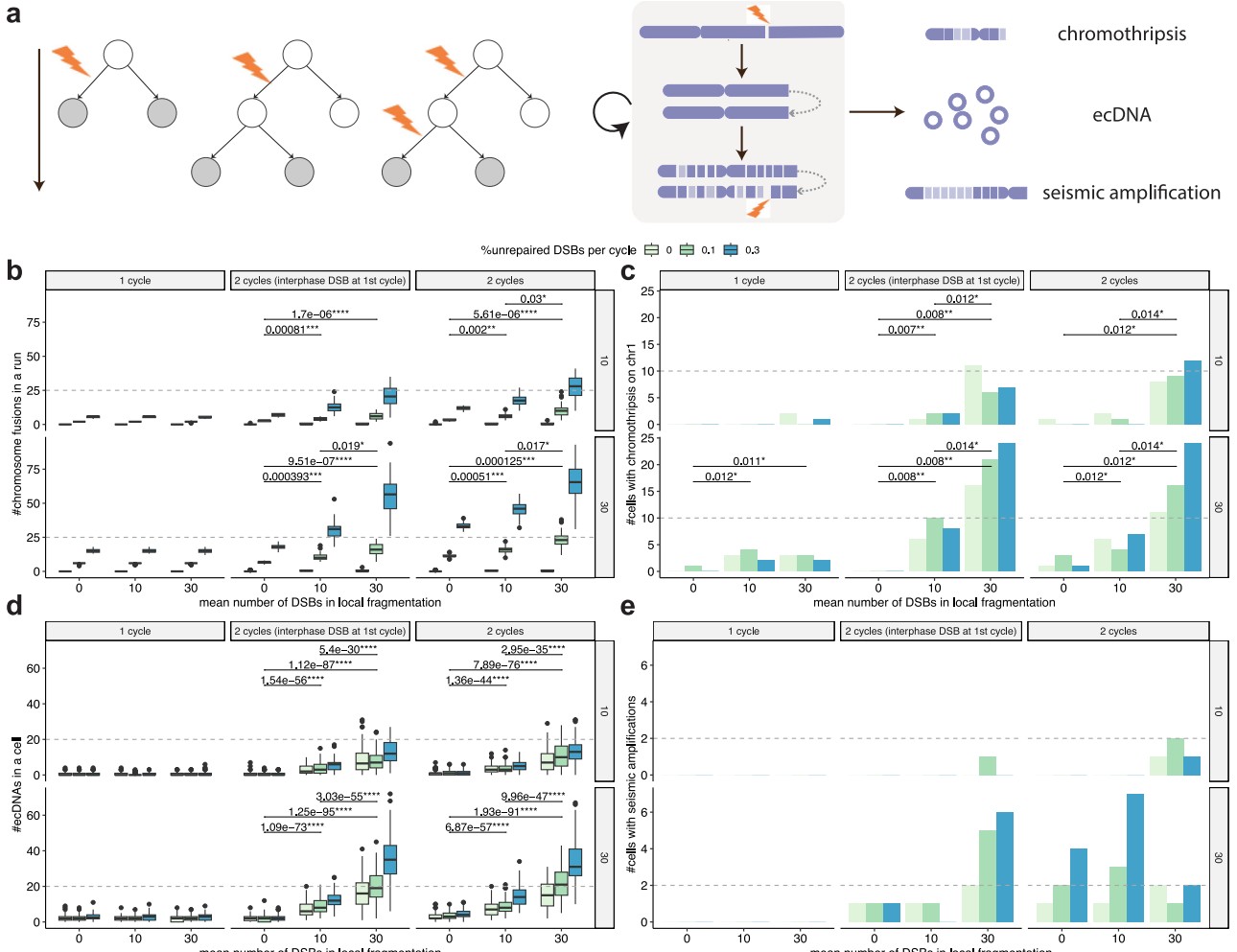

**Fig. 3 | Generation of complex structural variants (SVs) in one or two cell cycles.** **a** Overview of the model settings to validate the role of cell cycle in forming complex SVs. **b** The number of chromosome fusions in a run of simulation under different parameter settings in one or two cell cycles. Each box plot contains 50 data points, corresponding to 50 simulations under the same parameter setting. **c** The number of cells with chromothripsis on chr1 under different parameter settings in one or two cell cycles. **d** The number of extrachromosomal circular DNAs (ecDNAs) in a cell under different parameter settings in one or two cell cycles. Each box plot contains 100 data points, corresponding to 100 cells across 50 simulations under the same parameter setting. **e** The number of cells with seismic amplifications under different parameter settings in one or two cell cycles. The labels at the

right of each plot indicate DSB rates per cycle. For each run, we consider the two cells at the end of either the first or second cycle depending on the number of cycles (indicated by the grey cells in (**a**). The box plots show the median (centre), 1st (lower hinge), and 3rd (upper hinge) quartiles of the data; the whiskers extend to 1.5 times of the interquartile range (distance between the 1st and 3rd quartiles); data beyond the interquartile range are plotted individually. The significance levels of significant p-values from two-sided Wilcoxon tests are shown: * -- $p < 0.05$, ** -- $p < 0.01$, *** -- $p < 0.001$, **** -- $p < 0.0001$. The p-values were adjusted by Bonferroni correction for multiple pairwise tests. The two endpoints of the horizontal line below the p-value represent the two groups being compared. Source data are provided as a Source Data file.

likely due to the stringent criteria for detecting these events, which require at least six interleaved intra-chromosomal SVs and at least seven adjacent segments oscillating between two copy number states (Supplementary Fig. 11). When higher levels of local fragmentation were introduced, the involved chromosome was shattered into more pieces and hence generally led to significantly more BFB cycles, chromothripsis, and ecDNAs as well as slightly more seismic amplifications. This is consistent with the experimental results that most SVs occurred after local fragmentation[20]. With high-level local fragmentation, the numbers of BFB cycles and ecDNAs under two cell cycles were often significantly larger than those under one cell cycle, especially when DSB rate and percentage of unrepaired DSBs were also higher (Supplementary Fig. 12). However, the differences under the two cycles with or without consecutive interphase DSBs were relatively small, suggesting that the high probability of generating chromothripsis in two cell cycles with just a higher number of unrepaired interphase

DSBs and local fragmentation in one cell cycle. This may help to explain the wide prevalence of chromothripsis and ecDNAs in cancer genomes[27,28,47].

## Exploration of long-term evolutionary dynamics of ecDNA

Using simulations of larger cell populations under both neutral evolution and selection, our model can help to reveal the evolutionary dynamics of SVs in the long run. The ecDNA-based oncogene amplifications are common and associated with poor outcome across multiple cancer types[47], whose evolutionary dynamics has been recently well studied[48,49]. To compare with previous results, we continued our simulation of one unrepaired DSB with local fragmentation until 1000, 3000, and 5000 cells to see the long-term effect of selection on ecDNAs (Fig. 4a). As we simulated ecDNAs from DSB repair in the background of other SVs rather than in isolation, there may be different types of ecDNA with various numbers of copies in a cell. This is more

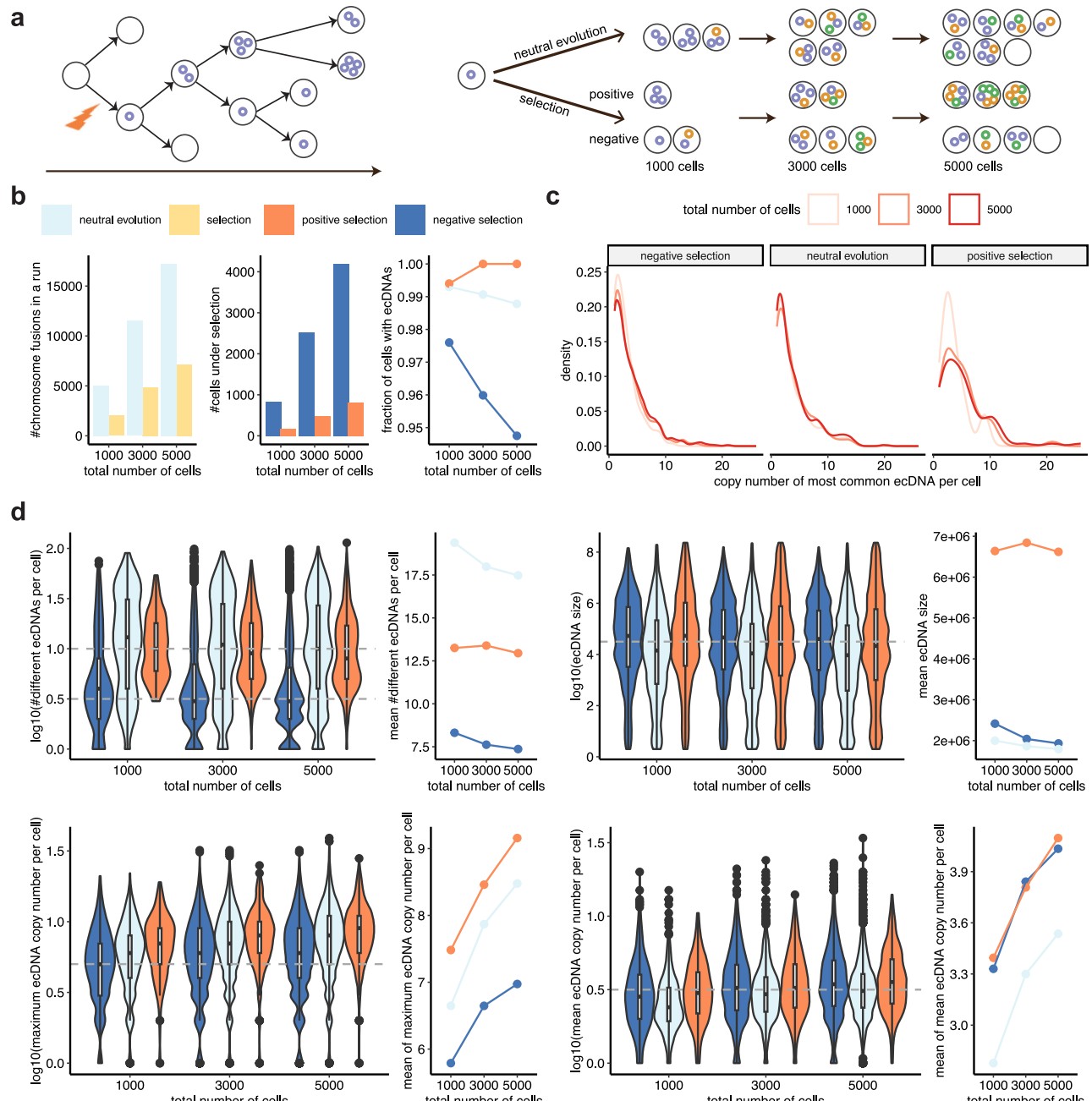

**Fig. 4 | Evolutionary dynamics of extrachromosomal circular DNAs (ecDNAs) under different selection forces over time. a** Overview of the model settings to simulate long-term evolution of ecDNAs from one unrepaired double-strand break (DSB) with local fragmentation under neutral evolution and selection. **b** The number of chromosome fusions, the number of cells under selection, and the fraction of cells with ecDNAs. **c** The distributions of copy numbers of the most common ecDNA type. **d** The distributions of the number, size, maximum copy number, and mean copy number of different ecDNAs per cell under different selection forces. The box plots show the median (centre), 1st (lower hinge), and 3rd (upper hinge) quartiles of the data; the whiskers extend to 1.5 times of the

interquartile range (distance between the 1st and 3rd quartiles); data beyond the interquartile range are plotted individually. When the total number of cells is 1000, 813, 993, and 166 cells contain 6760, 19,244, and 2200 ecDNAs under negative selection, neutral evolution, and positive selection, respectively. When the total number of cells is 3000, 2416, 2972, and 483 cells contain 18,411, 53,450, and 6468 ecDNAs under negative selection, neutral evolution, and positive selection, respectively. When the total number of cells is 5000, 3971, 4939, and 809 cells contain 29,245, 86,315, and 10,477 ecDNAs under negative selection, neutral evolution, and positive selection, respectively. Source data are provided as a Source Data file.

realistic than previous studies that focused solely on a single type of ecDNA associated with a specific oncogene[48,50]. Given recent findings from patient samples identifying multiple types of ecDNA[49], it is important to incorporate this diversity into the model. Our model of selection also realistically captures fitness changes of ecDNAs, which may undergo either positive or negative selection (Supplementary Fig. 13).

Our results support previous discoveries that ecDNAs show extreme variation across cells, with copy numbers and structural complexity progressively increasing over time[48,49] (Fig. 4b–d, Supplementary Fig. 13). The numbers of chromosome fusions under selection were much smaller than those under neutral evolution, which suggests that many fusions were disadvantageous, as more cells were under negative selection than positive selection (Fig. 4b). Although the

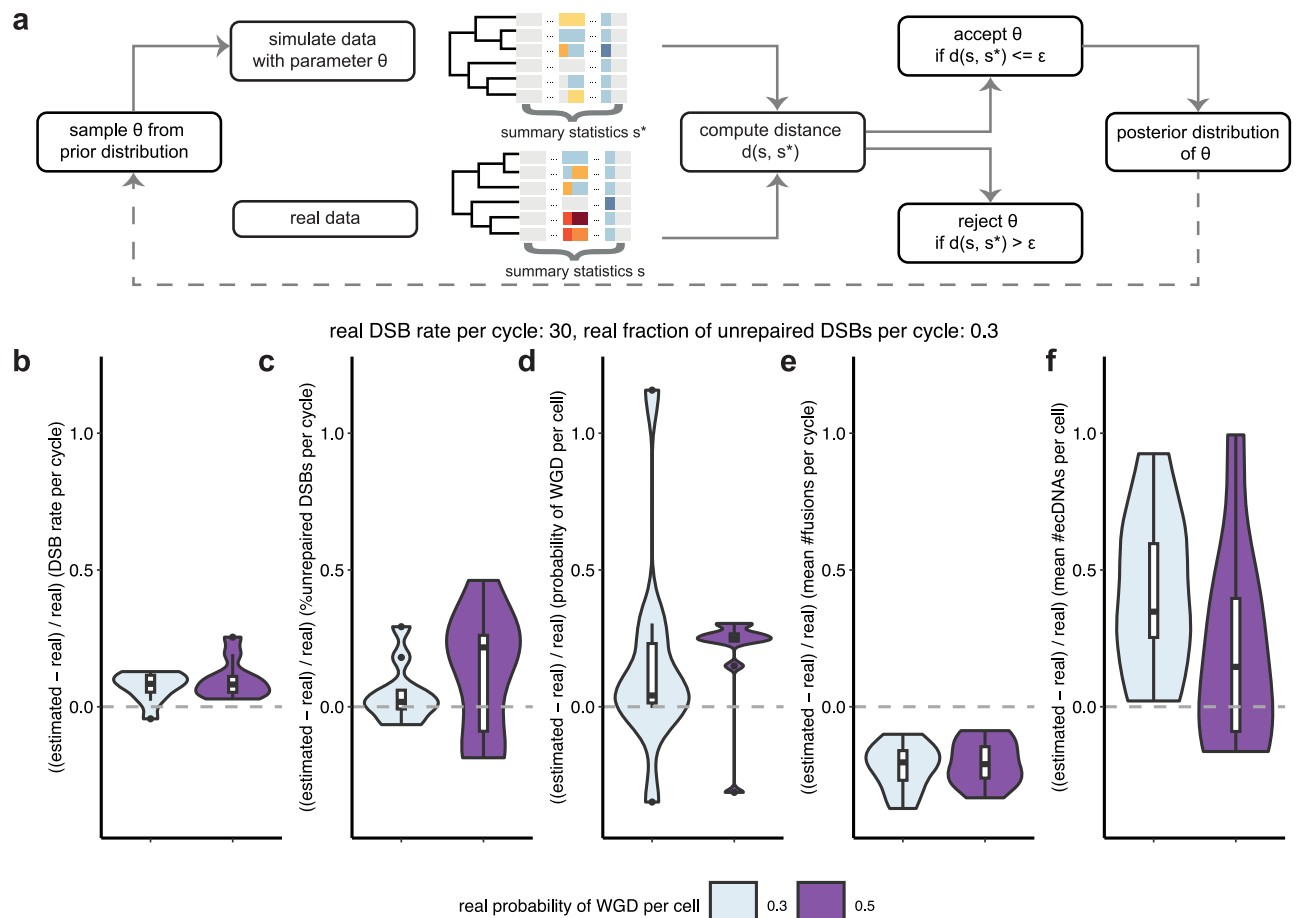

**Fig. 5 | Accuracy of parameter inference using approximate Bayesian computation sequential Monte Carlo (ABC SMC) on simulated data. a** Overview of ABC. The dashed arrow indicates that the posterior distribution contributes to the prior distribution in ABC SMC. **b** The proportion of differences between estimated and real double-strand break (DSB) rate per cycle. **c** The proportion of differences between estimated and real fraction of unrepaired DSBs per cycle. **d** The proportion of differences between estimated and real probability of whole genome doubling (WGD) per cell. **e** The proportion of differences between estimated and real mean number of chromosome fusions per cycle. **f** The proportion of differences between estimated and real number of extrachromosomal circular DNAs (ecDNAs) per cell. The box plots show the median (centre), 1st (lower hinge), and 3rd (upper hinge) quartiles of the data; the whiskers extend to 1.5 times of the interquartile range (distance between the 1st and 3rd quartiles); data beyond the interquartile range are plotted individually. There are 10 data points for each box plot, which represent the posterior means of 10 runs under the same parameter setting. Source data are provided as a Source Data file.

numbers of positively selected cells were relatively small, the fraction of cells with ecDNAs under positive selection kept increasing with rising fitness, whereas those under neutral evolution and negative selection kept decreasing (Fig. 4b, Supplementary Fig. 14). This pattern aligns with previous findings on the evolutionary dynamics of ecDNAs and indicates how different selection forces shape their prevalence[48]. For a specific type of ecDNA, the distribution of ecDNA copy numbers under strong positive selection was shown to shift towards higher copy numbers over time[48]. This trend is corroborated in the copy number distribution of the most common ecDNA type in our simulations (Fig. 4c). Additionally, the numbers of ecDNAs with different copy numbers over time showed similar distributions, with increasing maximum copy number (Supplementary Fig. 15). Although the sizes of ecDNAs and the mean ecDNA copy numbers per cell remained similar under both neutral evolution and selection, the numbers of ecDNA types and the maximum ecDNA copy numbers per cell were slightly higher under positive selection and neutral evolution (Fig. 4d). As cell population size increased, the mean ecDNA copy number per cell exhibited a small yet significant positive correlation with the selection coefficient, suggesting a tendency for cells with higher ecDNA copy numbers to be favoured by positive selection (Supplementary Fig. 16).

The slight changes of ecDNA sizes, moderate increases of mean copy numbers, and larger increases of maximum copy numbers over time (Fig. 4d) are consistent with previous discoveries obtained from patients with Barrett's oesophagus[49]. These patterns suggest ecDNA copy numbers can increase substantially during malignant progression, although ecDNA sizes may not show significant differences.

**Validation of parameter inference on simulated data**

To gain insight into the formation and evolution of rearranged genomes, we sought to develop a simulation-based inference approach using approximate Bayesian computation (ABC), which can be applied when the likelihood function is intractable[51] (Fig. 5a). We first came up with a set of informative and low-dimensional summary statistics for the simulated data based on domain knowledge, which aggregated over the complexity of genome rearrangement events (Methods, Supplementary Data 1). We inferred the posterior distributions of three key parameters in SV formation with ABC sequential Monte Carlo (SMC): DSB rate per cycle, fraction of unrepaired DSBs per cycle, and probability of WGD per cell, whereas the values of the other parameters of less interest were fixed to default values (Methods, Supplementary Fig. 17–18, Supplementary Table 1). We checked the predictive

power of our model by generating the posterior predictive distributions of two summary statistics arising in the simulation but often not directly observable from genome sequencing data: mean number of chromosome fusions per cycle and mean number of ecDNAs per cell (Methods). Over 90% of real values fell within two standard deviations of the posterior or posterior predicitive distributions for all five parameters (Supplementary Data 1). We calculated the fraction of summary statistics whose observed values fell within two standard deviations of the corresponding distributions, denoted as $P_w$. Over 74% of the datasets had $P_w = 1$ and only four datasets had $P_w < 0.9$, with the minimal value of $P_w$ being 0.83.

The results suggest that both DSB rate and fraction of unrepaired DSBs were accurately estimated especially when real values were higher, as indicated by the smaller differences between estimated and real values (Fig. 5b,c). Most real values fell within the main area of the posterior distributions (Supplementary Fig. 19a,b, 20a,b), although the distributions of posterior means suggest that they were slightly overestimated (Supplementary Fig. 21a,b). The posterior means of the probability of WGD were more overestimated (Fig. 5d, Supplementary Fig. 21c), and it is harder to accurately infer large values, as reflected by the wider posterior distributions (Supplementary Fig. 19c, 20c). This is because a large probability of WGD is very likely to give rise to repeated WGD whereas our simulation is constrained to at most one WGD per cell. The mean number of chromosome fusions per cycle was generally underestimated yet improved in precision with the increase of the real fraction of unrepaired DSBs, as chromosome fusions are mainly caused by broken ends (Fig. 5e, Supplementary Figs. 19d, 20d, 21d). The mean number of ecDNAs per cell was slightly overestimated and became more accurate when there were more ecDNAs in the data (Fig. 5f, Supplementary Figs. 19e, 20e, 21e). The obtained summary statistics from posterior predictive distributions were generally consistent with those observed in the simulated truth, whereas the deviances of some statistics were slightly larger, likely due to the difficulty in fitting a large number of summary statistics (Supplementary Fig. 22). The inferences using breakpoints sampled from a primary breast cancer patient showed similarly good performance (Methods, Supplementary Fig. 23). Taken together, these results show that our model-based simulation allows reliable Bayesian inference of important parameters in SV formation when there is sufficient information in the data and hence can be used to understand the processes underlying real genome rearrangements.

## Fitting the model to single-cell whole-genome sequencing data of epithelial cell lines and patient-derived xenografts

To demonstrate the utility of our model in learning SV generation mechanisms, we analysed 20 single-cell whole-genome sequencing datasets[43], where 8 datasets are from 184-hTERT mammary epithelial cell lines, and 12 datasets belong to FBI tumours from patient-derived xenografts (PDXs) of primary cancer patients, which are enriched in *CCNE1* amplifications (Methods, Fig. 6a, Supplementary Data 2). Since pseudobulk data were used for SV detection to achieve sufficient power for recovering breakpoints, SVs were reported for each individual clone in the datasets. Although this means we cannot apply our model to individual cells, we can still gain insights from clone-specific data where typical patterns of SVs can be identified. We simulated the branching process until it reached the number of clones in each dataset, and parameter interpretations were adjusted accordingly to clone expansion rather than cell division.

The results (Fig. 6b) were largely consistent with the observed SV patterns (Supplementary Figs. 24–28), with most datasets showing $P_w > 0.5$ except four datasets. The inferred parameters exhibited high heterogeneity across different datasets. Uniquely, our inference gave rise to posterior predictive distributions suggesting potential bounds on the unobserved numbers of BFB cycles (chromosome fusions) and ecDNAs. We excluded one poorly-fit dataset, DG1134, which has

approximately 27% tandem duplications (Methods, Supplementary Fig. 29a). On datasets with fewer polyploid clones, the inferred probability of WGD per cell was similar to the fraction of polyploid clones in each sample. In contrast, on datasets with more polyploid clones, the inferred value was generally smaller than the real fraction due to the constraint of at most one WGD in the simulation (Supplementary Fig. 29b). We detected few chromosome fusions and ecDNAs in the cell line datasets (Fig. 6b), which generally have fewer SVs than the FBI tumour PDX datasets. Our predictions indicate that SA1188, on average, has around one fusion and no ecDNA, which agrees with the typical patterns of BFBs detected on chr3q[43]. As it is still challenging to detect ecDNAs from single-cell data[52], our inferences may provide hints for further validations.

To understand the relationship among inferred parameters, we computed their posterior means for each dataset and plotted their pairwise correlations (Fig 6c–f, Supplementary Fig. 30). The posterior means of DSB rate, mean number of chromosome fusions, and mean number of ecDNAs significantly increased with probability of WGD (Fig. 6c–e). This is an emergent property of the dynamics since the model doesn't contain these dependencies, as demonstrated in the simulation results (Supplementary Figs. 31a–c, 32–33). Thus, we have shown that WGD can affect the generation of SVs through simple cellular processes. We also found significant correlations between chromosome fusions and ecDNAs in both single-cell whole-genome sequencing data and our simulated data (Fig. 6f, Supplementary Fig. 31d), suggesting that ecDNAs were mainly derived from BFB cycles and our model captured this feature.

The significant correlation between the probability of WGD and DSB rate is consistent with previous reports on increased levels of CNAs in cancers with WGD[11], elevated CNA rates after WGD in breast tumours[35], and rapid accumulation of arm-level CNAs triggered by WGD in Barrett's oesophagus and oesophageal adenocarcinoma[53]. There have been no prior results on the correlations between WGD and chromosome fusions. However, previous studies showed that WGD correlated with *CCNE1* amplifications[11], which is frequently targeted by BFB cycles[7]. This suggests a potential mechanistic connection between WGD and BFBs. Moreover, previous observations indicate that chromothripsis occurred after WGD in 74% of 194 cases where event orders can be distinguished[27]. This implies the correlation between WGD and ecDNAs, as chromothripsis is a major driver of ecDNA formation and amplification[15]. Additionally, a recent study indicates a positive association between WGD and ecDNAs in multiple tumour types[54]. Compared with cell line data, FBI tumours seem more likely to undergo WGD and have a much higher DSB rate, a slightly larger fraction of unrepaired DSBs, and many more BFB cycles and ecDNAs (Supplementary Fig. 34). The correlations of WGD with DSB rate, chromosome fusions, and ecDNAs may be explained by the enrichment of *CCNE1* amplification in FBI tumours. Given that *CCNE1* amplification is also associated with large tandem duplications[40], it seems instrumental in SV formation and CIN.

## Fitting the model to bulk whole-genome sequencing data of multiple cancer types

To demonstrate the broad applicability of our model across different tumour types, we applied it to deep bulk whole-genome sequencing data from the Pan-Cancer Analysis of Whole Genomes (PCAWG) Consortium[55] (Methods, Supplementary Table 1). Similar to single-cell data, parameter interpretations are based on clone expansion. In total, 111 PCAWG samples successfully finished ABC SMC, among which 29 samples exhibited WGD (Supplementary Figs. 35–36, Supplementary Data 3). Among these, 82 samples demonstrated a good fit, with most datasets showing $P_w > 0.5$ except 10 datasets (Supplementary Figs. 37–38), while the remaining 29 samples (without WGD) were poorly fitted (Supplementary Fig. 39). Compared to well-fit samples, poorly-fit samples have significantly more samples with over 25%

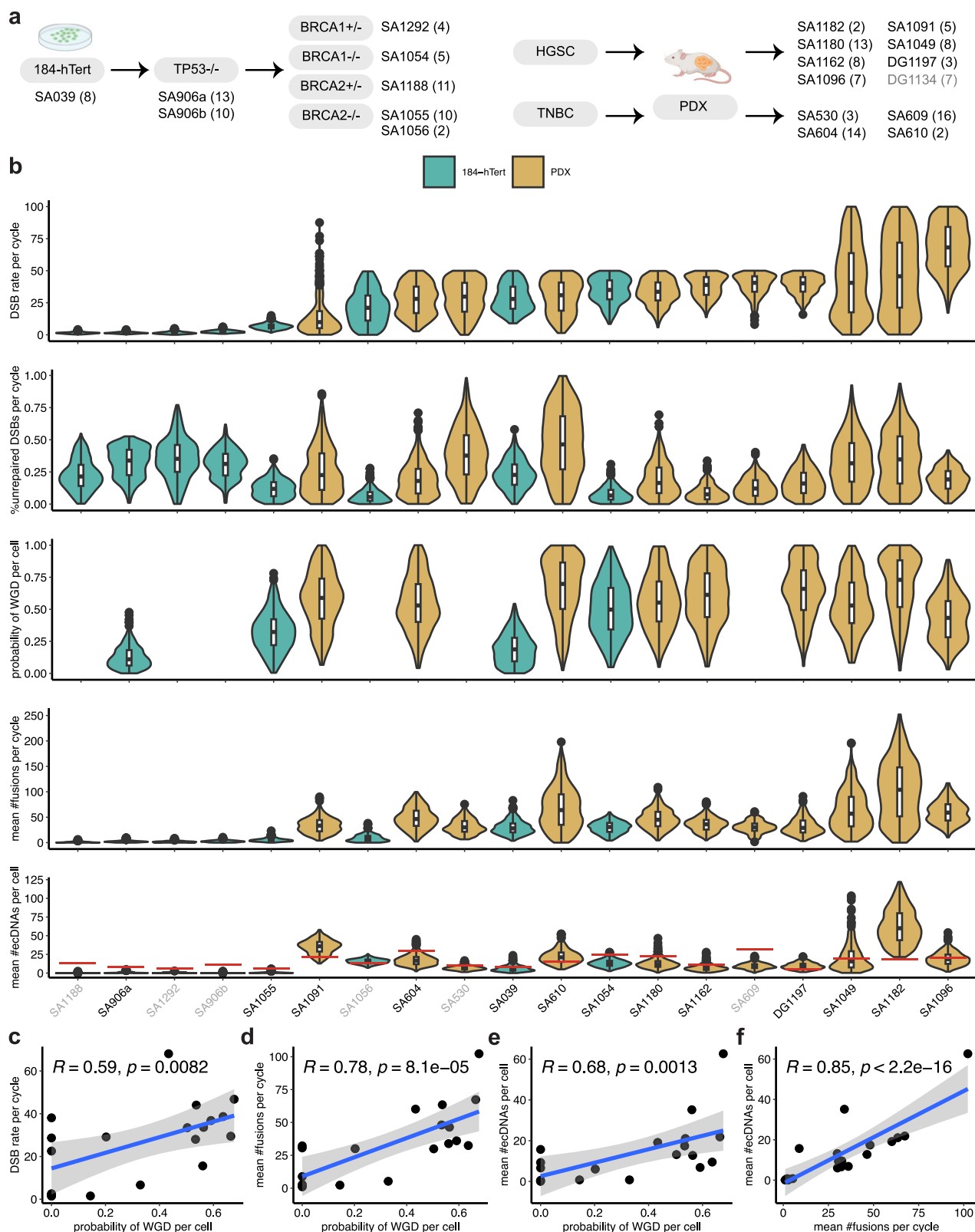

tandem duplications (two-sided chi-squared test, *p*-value = 0.003422) and significantly fewer samples with over 25% inversions (two-sided chi-squared test, *p*-value = 0.04373). The distributions of inferred parameters for the 82 well-fit samples across ten cancer types demonstrated high heterogeneity (Fig. 7, Supplementary Fig. 40). The ranking of cancer types by posterior means of DSB rates significantly correlated with the average number of SVs per sample across 1551

PCAWG samples from the same cancer types (Kendall's rank correlation coefficient 0.78, *p*-value 0.0009463). Because each sample is classified as either having WGD or not, the fraction of clones with WGD is always 1, suggesting the inferred probabilities of WGD, with posterior means around 0.6, are reasonable. The posterior means of DSB rate, mean number of chromosome fusions, and mean number of ecDNAs significantly increased with the inferred probability of WGD,

**Fig. 6 | Results of fitting the cell-cycle model to single-cell whole-genome sequencing data using approximate Bayesian computation sequential Monte Carlo (ABC SMC). a** Overview of 20 single-cell datasets used for analysis. The number within the parentheses indicates the number of clones in a dataset. The grey colour for DG1134 indicates that it was poorly fit. **b** The posterior distributions of three inferred parameters and the posterior predictive distributions of the unobserved numbers of chromosome fusions and extrachromosomal circular DNAs (ecDNAs) for 19 well-fit datasets. The names of datasets without whole genome doubling (WGD) are shown in grey. The red bar in the last group of violin plots indicates the number of ecDNAs from database that overlap with structural variants (SVs). The box plots show the median (centre), 1st (lower hinge), and 3rd (upper hinge) quartiles of the data; the whiskers extend to 1.5 times of the interquartile range (distance between the 1st and 3rd quartiles); data beyond the interquartile range are plotted individually. Each box plot contains 500 data points, which represent the posterior samples. **c** The correlation between the inferred probability of WGD per cell and DSB rate per cycle. **d** The correlation between the inferred mean number of chromosome fusions per cycle and probability of WGD per cell. **e** The correlation between the inferred mean number of ecDNAs per cell and probability of WGD per cell. **f** The correlation between the inferred mean number of ecDNAs per cell and mean number of chromosome fusions per cycle. In (**c**–**f**) the Spearman correlation coefficient and corresponding two-sided *p*-value are shown for each plot, with the shaded area showing the 95% confidence interval of linear regression and each point representing the posterior mean. Source data are provided as a Source Data file.

along with a significant correlation between chromosome fusions and ecDNAs (Supplementary Fig. 41a). Analysis of 56 samples with inversion enrichment (fraction exceeding 25%) (Supplementary Fig. 41b) and 15 samples with both *CCNE1* amplifications and inversion enrichment (Supplementary Fig. 41c) also revealed similar significant correlations.

Although ecDNAs, BFBs, and chromothripsis were previously detected in 57 of the well-fit samples[47], they were mostly detected as present or absent. Only three samples (SA505351, SA505919, and SA505711) have more than one ecDNA detected, suggesting the real number of ecDNAs in a sample is likely higher than previously reported. For example, sample SA557416 has 12 chromothripsis events detected but only one ecDNA reported, suggesting the possible presence of undetected ecDNAs. Among the 25 samples without reported ecDNAs, BFBs, or chromothripsis, SA501627 and SA533743 have heavily rearranged SVs detected. We used the eccDNAdb database, which includes ecDNAs from tumour samples derived from patient tissues, PDXs, and cancer cell lines[56], to identify ecDNAs overlapping with SVs in each sample. Only three samples (SA517374, SA530582, and SA541782) lack SV-overlapping ecDNAs, despite the detection of chromothripsis events in these samples.

The notably higher posterior means of ecDNAs and chromosome fusions compared to the known numbers of events suggest potential underestimations of complex SVs in cancer genomes (Fig. 8). For comparison, we included the results from single-cell whole-genome sequencing data (Fig. 8a), which showed a linear relationship between the inferred number of ecDNAs and the number of SV-overlapping ecDNAs, similar to the pattern observed in PCAWG data (Fig. 8b). These monotonically increasing relationships suggest our inferences align with the potential scale of the real number of ecDNAs in the datasets. The known number of chromothripsis events exhibited a significant correlation with the posterior means of ecDNAs (Fig. 8c), whereas it showed an insignificant correlation with the posterior means of chromosome fusions (Fig. 8d). These correlations support the key role of chromothripsis in driving ecDNA formation, with BFB cycles sometimes contributing to the process[15,46]. Due to the limited detection of ecDNAs and BFBs, no meaningful patterns emerged from comparisons between the known and inferred numbers of ecDNAs (Fig. 8e), as well as BFBs or chromosome fusions (Fig. 8f). However, this is likely to improve with more accurate detection methods, such as long-read sequencing and associated SV detection techniques.

The analysis of mutational, copy number, and SV signatures revealed different activities of underlying mutational processes in the well-fit samples. Among the known mutational signatures in PCAWG datasets, those present in more than 10% of the well-fit samples predominantly correlated with age (SBS1, SBS5, SBS40, DBS2, DBS4, ID1, ID2, ID5, and ID8) (Supplementary Fig. 42). SBS1, ID1, and ID2 were simultaneously present in 67 samples and have been proposed to arise during DNA replication in mitosis[57]. Copy number signatures of 50 well-fit samples, extracted from[30], suggest that CX1 and CX12 were significantly dominant in SVs arising from NHEJ (Supplementary Fig. 43). Additionally, two SV signatures from the Catalogue Of Somatic

Mutations In Cancer (COSMIC) database and two de novo SV signatures, which are enriched with non-clustered translocations or inversions, were present in over half of the well-fit samples (Supplementary Fig. 44, Supplementary Data 4).

## Discussion

In summary, we proposed a quantitative cell-cycle model for DSB repair by end joining and replication. This model generated a wide spectrum of patterns caused by CIN at the whole genome level. Our model incorporates DNA damage along with erroneous repair and replication processes, which has not been attempted previously. This approach provides a natural way to integrate CNAs and SVs, which have often been analysed independently despite being intrinsically coupled by unifying mutational processes[7]. Through simulations under diverse scenarios, we showed how various chromosomal alterations developed and correlated in the short or long term. Our simulations recaptured genomic patterns of several complex SVs, revealed factors affecting the formation of catastrophic SVs such as chromothripsis, and illustrated the evolutionary dynamics of ecDNAs. Since it is hard to directly observe DSBs and their outcomes in practice, our in silico simulations make it possible to investigate the intermediate SV formation steps that are otherwise unobservable, allowing us to deconvolve the intricacies of complex SVs and explore their long-term evolutionary dynamics. By fitting to simulated data and single-cell/bulk whole-genome sequencing data with ABC, we demonstrated the power of our model in recovering key parameters in SV generation and revealing previously unrecognised correlations among SVs. The generation of multiple SVs initiated by BFB in the same DSB repair process provides support for combining chemotherapy/radiotherapy and DNA repair inhibitors in cancer treatment[15]. The critical role of WGD and *CCNE1* amplification uncovered by our model from single-cell/bulk whole-genome sequencing data also provides evidence for targeting *KIF18A* that is required for viability of cells with WGD or *CCNE1* in the clinic[58,59].

Most evolutionary inferences require a mutational model, such as an appropriate null model to detect driver SVs[60], and our model may serve as such a null model for hypothesis testing and model selection on related problems. By disentangling functional and non-functional SVs under selection, our model may help with distinguishing the contribution of mutation and selection in tumours driven by CIN and SVs[5]. As more SVs are anticipated to be discovered with long-read sequencing and better detection methods[10], our model will provide a useful tool to facilitate understanding of mechanisms leading to these SVs. The model may also be extended to fit other types of data such as single-cell data from experimental studies observing cell division in vitro when both CNAs and SVs can be detected in each cell[33].

To get a tractable model of complicated cell processes, we made several simplifying assumptions and hence there are a few limitations. First, we excluded biological constraints on breakpoint locations and timing when introducing DSBs and contact probabilities when repairing DSBs. These constraints may be integrated in the future to increase

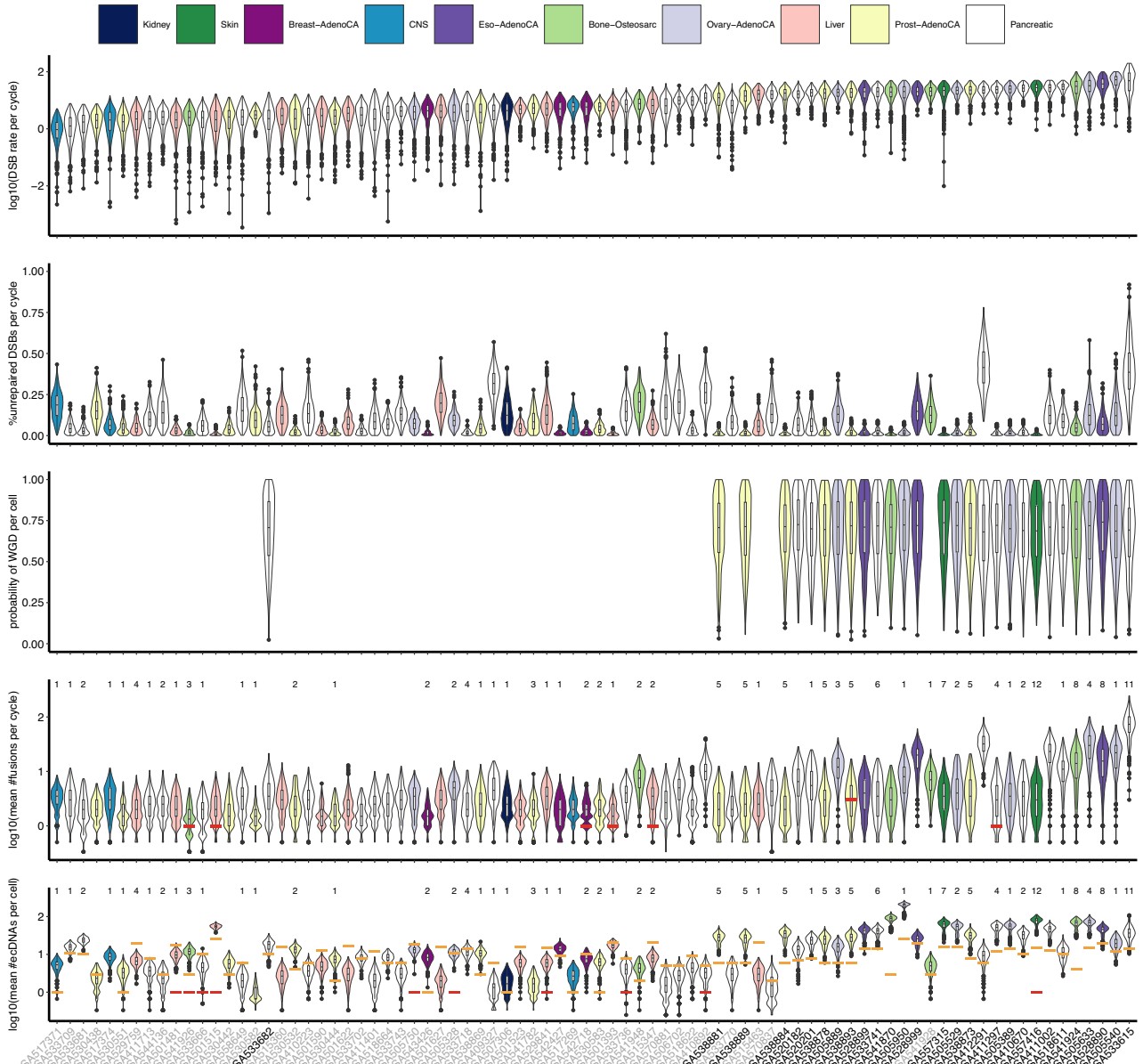

**Fig. 7 | Results of fitting the cell-cycle model to bulk whole-genome sequencing data using approximate Bayesian computation sequential Monte Carlo (ABC SMC).** The five groups of violin plots show the posterior distributions of three inferred parameters and the posterior predictive distributions of the unobserved numbers of chromosome fusions and extrachromosomal circular DNAs (ecDNAs) for 82 well-fit bulk whole-genome sequencing datasets. The names of datasets without whole genome doubling (WGD) are shown in grey. The red bar in the last two groups of violin plots indicates the number of detected breakage-fusion-bridges (BFBs) or ecDNAs. The orange bar in the last group of violin plots indicates

the number of ecDNAs from database that overlap with structural variants (SVs). The numbers above the two bottom groups of violin plots represent the numbers of detected chromothripsis. The box plots show the median (centre), 1st (lower hinge), and 3rd (upper hinge) quartiles of the data; the whiskers extend to 1.5 times of the interquartile range (distance between the 1st and 3rd quartiles); data beyond the interquartile range are plotted individually. Each box plot contains 500 data points, which represent the posterior samples. Source data are provided as a Source Data file.

the accuracy of our model, such as assigning higher DSB probabilities to fragile sites or DSB hotspots, adding replication-associated DSBs in the $S$ phase, incorporating 3D nuclear distances (including chromatin organisation and state) for more accurate contact probabilities, and constraining breakpoint rejoining with sequence homology as required by different repair pathways[24,60]. Note that a large fraction of DSBs still occur at fragile sites in our simulations, although DSBs were introduced randomly or sampled from known breakpoints (Supplementary Fig. 45). Second, we simplified DNA repair and replication by directly rejoining two breakpoints and correctly duplicating all the DNAs, which prevents modelling of certain SVs resulting from HRD or

complicated replication errors[25,39]. To expand the applicability of our model to more SV types, more details such as replication-based template switching need to be incorporated. Third, the model of selection is based only on large-scale CNAs using the density of OGs and TSGs, which can be improved by considering other factors, such as SV size[34], gene density[61], maximum ecDNA copy number advantageous for cell survival[50], fitness cost of unrepaired DSBs or balanced SVs[60], and tissue-specificity[62], to better quantify selection strengths on SVs. Lastly, due to computational complexity, our simulations are not efficient enough to allow ABC on datasets with a large number of cells or clones, which need to be further optimised.

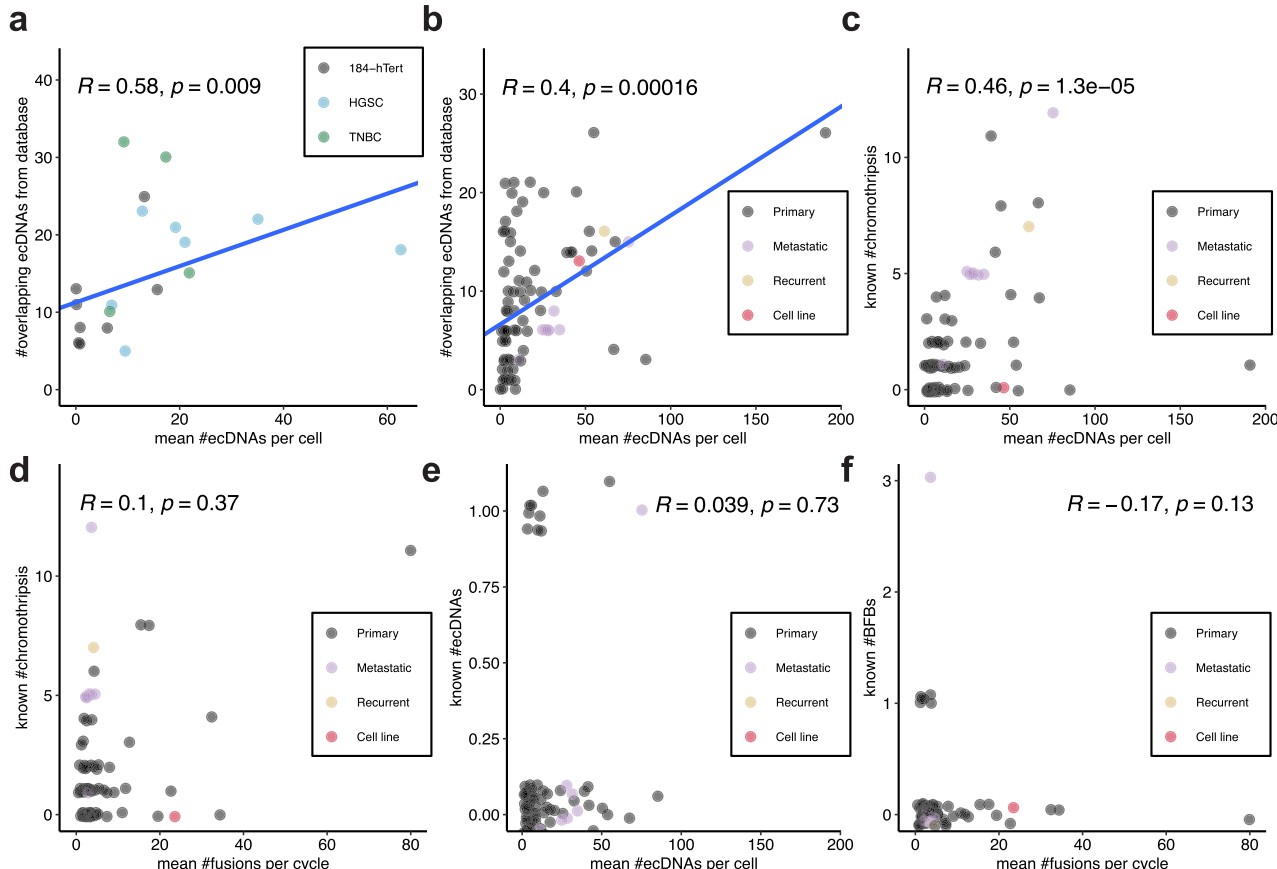

**Fig. 8 | Relationships between inferred values and known numbers of events.**
**a** The relationship between inferred posterior means of extrachromosomal circular DNAs (ecDNAs) and numbers of ecDNAs from database that overlap with structural variants (SVs) for 19 well-fit single-cell whole-genome sequencing datasets. **b** The relationship between inferred posterior means of ecDNAs and numbers of SV-overlapping ecDNAs from database for 82 well-fit bulk whole-genome sequencing datasets. **c** The relationship between inferred posterior means of ecDNAs and known numbers of chromothripsis for 82 well-fit bulk whole-genome sequencing datasets. **d** The relationship between inferred posterior means of chromosome fusions and known numbers of chromothripsis for 82 well-fit bulk whole-genome sequencing datasets. **e** The relationship between inferred posterior means and known mean numbers of ecDNAs for 82 well-fit bulk whole-genome sequencing datasets. **f** The relationship between inferred posterior means and known numbers of breakage-fusion-bridges (BFBs) or chromosome fusions for 82 well-fit bulk whole-genome sequencing datasets. In (**a**, **b**) the blue lines were derived from robust linear regressions. The Spearman correlation coefficient and corresponding two-sided *p*-value are shown for each plot. Jittering with both height and width set to 0.1 are applied for better visualisation of overlapping data points. Source data are provided as a Source Data file.

In conclusion, our model contributes to the development of a unified framework to investigate the biological processes underlying various patterns of CIN and to the understanding of the relationship among related SV types. We have been able to quantify CIN and SV generation in single-cell and bulk whole-genome sequencing data, which has not been done before. Although these phenomena have been separately studied, we have demonstrated the close relationship of BFB cycle, chromothripsis, and ecDNA all through the processes of erroneous end joining and replication across cell cycles. Our framework provides a foundational null model for SV formation and can be used to disentangle functional and non-functional heterogeneity to guide cancer treatment development. Leveraging rapidly advancing deep learning techniques[63], we will improve the efficiency of our simulation to make it applicable to larger-scale data.

## Methods
### Stochastic cell-cycle model of SV generation
We simulated cell growth from a single cell until reaching $N$ cells with the rejection-kinetic Monte Carlo algorithm[64]. To distinguish SVs generated in a few cell cycles and SVs generated gradually over all the cell cycles, we allowed the setting of the maximum number of cycles with DSBs ($n_d$). To be realistic and facilitate fitting the model to

genomic data, we simulated the diploid genome with the coordinates of 22 autosomes of the reference human genome (GRCh38/hg38 by default).

The cell-cycle repair dynamics were inspired by a model of dose-dependent DSBs and rejoining after ionising radiation[65]. In $G_1$, we introduced $n$ (either fixed or variable following Poison distribution with a pre-specified mean value $r$ per cell cycle) DSBs on randomly selected chromosomes under infinite sites assumption where each DSB hits a different position. Although NHEJ occurs throughout the cell cycle, it mainly occurs in $G_1$ due to the absence of homologous sequences, such as sister chromatids, which are required for HR[66]. Additionally, replication-associated DSBs occurring in the S phase are more commonly repaired via HR[66]. Therefore, we primarily introduced DSBs in $G_1$ and did not introduce additional DSBs in the other phase, except in $M$. For simplicity, we excluded DSBs within telomeres and centromeres. In the analysis of chromothripsis in 634 adult tumours across 28 cancer types, the telomere and centromere regions were affected by chromothripsis in around 36% and 55% of the cases respectively[28], and hence we can still simulate a large proportion of realistic SVs. Each chromosome $c$ has a probability $p_c$ of generating a DSB, where $1 \le c \le 22$ and $\sum p_c = 1$. By default, each chromosome had the same probability of having DSBs. To facilitate the generation of

chromothripsis events which often occur on one chromosome, we allowed different probabilities of DSBs on different chromosomes. This will not only generate clustered breakpoints enriched on the chromosome with the largest probability but also cause chromosome fusions derived from rejoining of inter-chromosomal breakpoints. When trying to bias towards one chromosome, we set its probability to $\frac{2}{3}$ and the other chromosomes to have equal probability whose sum was $\frac{1}{3}$. Once a chromosome was chosen for DSB, we randomly selected a position in a genomic interval on one homologue. As some random breakpoints may be less likely to occur in reality, we also allowed random sampling of DSBs without replacement from breakpoints extracted from detected SVs, weighted by their frequencies if available. To simulate catastrophic events such as chromothripsis, we allowed the introduction of many DSBs in the first $n_d + 1$ cycle(s) and no DSBs in the subsequent cycles. When the number of divisions for a cell exceeds $n_d$ (default zero, namely DSBs in $G_1$ only occur at the first division), no more DSBs will be introduced. When $n_d + 1$ equals or exceeds the maximum number of cell cycles, DSBs will be introduced during $G_1$ in every cell cycle.

We repaired a fraction $(1-f_u)$ of either all the DSBs in the cell (default) or the new DSBs introduced in the current cell cycle with different options, where the unrepaired DSBs may be repaired in the subsequent cell cycles or not. One breakpoint will be left unrepaired when there is an odd number of breakpoints. Note that unrepaired DSBs may lead to chromosome fusion and differ from misrepaired DSBs. The first repair option is randomly joining two breakpoints, where $\frac{1}{n}$ of them may be faithfully rejoined on average. However, the probability of successful DSB repair often depends on the distance of the two breakpoints in the genome or the nucleus, and spatially close DSBs are more likely to be rejoined and form SVs[60,67]. Therefore, the contact probability of two breakpoints should decrease with their genomic distance in a single chromosome or their spatial distance across chromosomes. Namely, breakpoints further away from each other on the same chromosome and breakpoints on different chromosomes are less likely to join. To be more realistic, we introduced another repair option (default), where breakpoints are rejoined based on the above observations. We first selected one breakpoint $b_1$ and then chose the other breakpoint $b_2$ according to the reciprocal of its distance to $b_1$ with discrete distribution. The distance of $b_1$ and $b_2$ was defined as $|b_1 - b_2|$ when they are on the same chromosome and $10^9$ when they are on different chromosomes. As the two breakpoints introduced by the same DSB have distance one, we assigned their contact probability to be $p_r$ to control the proportion of unfaithful rejoinings[65]. The repair was based on genomic distance, with the default probability of correctly repairing a DSB set to $p_r = 0$, as the correctly repaired DSBs are equivalent to no DBSs being introduced. Circular DNAs with or without centromere(s), namely centric or acentric rings[65], linear chromosomes without centromere(s), and chromosome fusions may be generated due to random rejoining. For example, an isolated acentric piece without telomeres forms a circle by a new variant adjacency connecting the two ends.

In $S$ and $G_2$, we replicated the whole genome and connected broken chromosomes with lost telomeres to form fusions and dicentric chromosomes, which imitates telomere crisis or shortening[68]. For circular chromosomes and complete chromosomes with both telomeres (e.g., pink chromosome in Fig. 1), we made an exact copy. Due to unrepaired breakpoints, some chromosomes may lose one or both telomeres. For a non-circular chromosome lacking one telomere or both telomeres (e.g., purple and green chromosomes in Fig. 1), we linked it to its copy, imitating sister chromatid fusion. The genomic segment lacking both telomeres will form a circular DNA. We consider the genomic segment lacking both telomeres and a centromere as an ecDNA, which can be formed either during $G_1$ or during chromosome fusion.

In $M$, we distributed the replicated chromosomes into two daughter cells according to the number of centromeres $N_c$ in each

chromosome. If $N_c = 1$, the chromosome was inherited by either daughter cell. If $N_c = 0$, the chromosome was randomly distributed to a daughter cell. As a result, one daughter cell may have two copies of this chromosome and the other daughter cell may lose it completely, leading to reciprocal gain and loss. If $N_c > 1$, the chromosome was subject to DSB, imitating chromosome segregation errors such as bridge breakage. Previous experiments suggest that direct bridge breakage generated either a simple break or local fragmentation which may affect two or more chromosomes[20]. We simulated a simple break by breaking the chromosome at feasible breakpoints according to the locations of all the centromeres in the fused chromosome to ensure that each derivative chromosome had a single centromere and was allocated to a different daughter cell when $N_c = 2$ and to a random daughter cell when $N_c > 2$. We simulated local fragmentation by breaking the derivative chromosome with one centromere after a simple break into $n_l$ pieces, where $n_l$ followed a Poison distribution with a pre-specified mean $m_l$, and each piece was randomly allocated to a daughter cell. When $n_l > 0$, we introduced local fragmentation with a probability of 0.5 to avoid too much shattering. We also simulated WGD in each cell with probability $p_w$, assuming at most one WGD in a cell, by allocating all the chromosomes to one daughter cell and letting the other daughter cell die.

We used OG-TSG score, derived from mutational profiles and correlated with amplification frequency, to measure cell fitness based on the density and potency of OGs and TSGs overlapping with chromosome-level or arm-level (default) CNAs[2,41]. We computed the survival probability of each cell with the formula[69]:

$$P_{survival} = \exp\left(d \sum_{i=1}^{m} c_i s_i + c\right), \quad (1)$$

where $d = 0.00039047$, $c = -0.036132164$, $m$ is the number of chromosomes or arms, $c_i$ is the average total copy number of chromosome or arm $i$, and $s_i$ is the corresponding OG-TSG score. The values for $d$, $c$, and $s_i$ are the same as those in ref. 69, which have been used in ref. 2. When the genome is normal, $P_{survival} = 0.9689715$. Assuming exponential growth of cells, the population size at time $t$ is

$$N(t) = R^t = e^{ln(R)t} = e^{(b-d)t}, \quad (2)$$

where $R$ is the average number of offspring of a cell. Suppose $\alpha = P_{survival}^{S}$ is the probability for a daughter cell to survive, where $S > 0$ is introduced to control the magnitude of selective pressure[61], then the offspring distribution is $\mathbf{P} = (p_0, p_1, p_2)$, where $p_0 = (1-\alpha)^2$, $p_1 = \alpha(1-\alpha)$, and $p_2 = \alpha^2$. Therefore,

$$R = E(\mathbf{P}) = p_1 + 2p_2 = \alpha(1-\alpha) + 2\alpha^2 = \alpha(1+\alpha). \quad (3)$$

Then we can get the relationship between the survival probability of a cell and its birth/death rate:

$$b - d = ln(R) = ln(\alpha(1+\alpha)). \quad (4)$$

Assuming the birth and death rate of the normal cell and its daughter cell are $b_0$, $d_0$, $b_1$, and $d_1$ respectively, then

$$\frac{b_1 - d_1}{b_0 - d_0} = \frac{ln(\alpha_1(1+\alpha_1))}{ln(\alpha_0(1+\alpha_0))} = 1 + s, \quad (5)$$

where $s$ is the selection coefficient measuring the strength of natural selection on the daughter cell. When $s = 0$, it means neutral evolution and the daughter cell has the same birth/death rate as the parent cell. When $s > 0$, it means the daughter cell is positively selected and has either a higher birth rate or a lower death rate. When $s < 0$, it means the daughter cell is negatively selected and has either a lower birth rate or

a higher death rate. We then derived the value of $s$ from $\alpha_0$ and $\alpha_1$ and updated the birth/death rate of the daughter cell accordingly. Note that when $S$ is no larger than 15, the larger the $S$, the stronger the selection is.

The programme generated detailed output files for inference and validation, which include SVs and haplotype-specific CNAs in each cell, the derivative genome represented by a list of connected intervals for each cell, and the cell lineage relationship generated in the branching process. The CNAs were reported at both segment level of variable size and bin level of fixed size with pre-specified locations. We obtained haplotype-specific copy numbers for a segment by merging consecutive intervals with the same copy number on a homologue of a chromosome. To get haplotype-specific copy numbers for non-interval adjacencies, we traversed all the adjacencies and grouped them by positions and orientations. To get bin-level copy numbers which are often detected in bins of fixed size such as 500 Kbp, we allocated each segment to bins of size $B$. We computed the bin IDs, $\mathbf{b_i}$, for a segment $i$ with start position $s_i$ and end position $e_i$, by computing the start bin ID as $b_i^1 = \frac{s_i}{B}$ and the end bin ID as $b_i^m = \frac{e_i}{B}$, where $m = \frac{e_i - s_i + 1}{B} + 1$ is the number of bins covering the segment. Since other segments may overlap with the bins at both ends of segment $i$, the contribution of segment $i$ to the copy number of these bins is weighted by the overlap size.

We computed multiple summary statistics for each run of the simulation (Supplementary Data 1), among which the fraction of cells with WGD, the fraction of different types of SVs (deletion, duplication, inversions, and intra-chromosomal SVs), the frequency distribution of breakpoints present in different numbers of cells, the percentage of genome altered (PGA), and the mean and standard deviation of pairwise divergence were used for inference with ABC. PGA and the mean and standard deviation of pairwise divergence were computed based on total and haplotype-specific copy numbers, respectively. A bin is altered if its copy number is different from the ploidy, which is two for normal cells and four for cells with WGD when considering total copy number, and one for normal cells and two for cells with WGD when considering haplotype-specific copy number. We defined PGA as the percentage of bins that are altered (in any cell) from the ploidy across all cells and pairwise divergence between two cells as the proportion of bins with different copy numbers that are altered from the ploidy.

### Graph representation of the genome

We defined a breakpoint $b$ as the position on a chromosome $c$ ($1 \leq c \leq 22$) of size $l$ ($0 < b < l$), with a label $H \in \{A, B\}$, where $A$ and $B$ represent the two homologues of $c$ that are inherited from each parent. A DSB at position $j$ of a chromosome introduced two breakpoints $j$ and $j + 1$, each of which was represented as a node in the graph. Each breakpoint divided a segment $[i, k]$ into two intervals $[i, j]$ and $[j + 1, k]$. Following conventions[27], the orientation of the left breakpoint $j$ was denoted by +/HEAD (the right side or head of interval $[i, j]$), whereas the orientation of the right breakpoint $j + 1$ was denoted by -/TAIL (the left side or tail of interval $[j + 1, k]$). Most breakpoints were introduced in $G_1$ as a result of interphase DSBs, and the remaining breakpoints were generated in $M$ due to bridge breakage. The adjacency orientations on the same chromosome, determined based on the involved breakpoints, indicate simple SV-like patterns of four types: tandem duplication-like ($-/+$), deletion-like ($+/-$), head-to-head inversion ($+/+$), and tail-to-tail inversion ($-/-$), whereas adjacencies across two different chromosomes indicate inter-chromosomal SVs.

The genome was represented as a set of paths in the graph $G$, where a path is a walk in $G$ alternating between interval and reference/variant adjacency edges. A path represents a derivative chromosome and may contain a varying number of centromeres and telomeres (no telomere, left/right telomere, and both telomeres). We represented a circular path by adding one more adjacency between the two ends of the corresponding linear path. We did a walk along the graph at the

end of $G_1$ to get all the paths in the genome before DNA replication. Among the adjacencies introduced in one cell cycle, most occurred during breakpoint rejoining in $G_1$ and the remaining variant adjacencies happened during chromosome fusion in $S$ to represent head-to-head and tail-to-tail inversions. In $M$, due to bridge breakage, some genomic intervals may get broken and lead to new adjacencies in $G_1$ of the next cell cycle.

### Generation and analysis of simulated data

To reduce visualisation burden and get a variety of SVs in Fig. 1c, we only introduced five DSBs at each cell cycle, which were biased towards chr1.

To learn the effect of a single BFB with simple breaks, we introduced one DSB on a random chromosome at the first cell cycle and left it unrepaired until 100 cells under both neutral evolution and selection (selection strength $S = 5$). To further show our model can simulate more complex patterns such as chromothripsis under local fragmentation, we set the mean number of DSBs during local fragmentation to 50 to get more breakpoints in a short time. We introduced two DSBs at the first cell cycle with 50% of DSBs correctly repaired ($p_r = 0.5$) at each cell cycle, which left one unrepaired DSB after the first cell cycle. This unrepaired DSB led to ongoing BFB cycles, where the resultant bridges underwent simple breaks or local fragmentation randomly in each cell cycle. We used the same seed for simulations under neutral evolution and selection so that the resultant differences were due to selection rather than stochasticity. To examine the relationship between chromothripsis and ecDNAs, we repeated the simulation with local fragmentation (Supplementary Fig. 1) 50 times, each with a different random seed. To explore the interaction among WGD, chromothripsis, and ecDNAs, we performed additional simulations with increasing probabilities of WGD (0.1, 0.122, and 0.124) while keeping all other parameters the same as those used for the simulation with local fragmentation. We also repeated the simulations 50 times when the probability of WGD was 0.1 and 0.122, respectively.

To determine the minimum number of misrepaired DSBs required to generate chromoplexy, we conducted simulations with progressively fewer misrepaired DSBs per cycle (5, 2, and 1) on random chromosomes in each cell cycle, with 10 repeats for each parameter setting.

For simulations on the formation of chromothripsis, we simulated data with DSB rate per cycle $r \in \{10, 30\}$, fraction of unrepaired DSBs per cycle $f_u \in \{0, 0.1, 0.3\}$, and mean number of DSBs per fragmentation) $n_l \in \{0, 10, 30\}$ (50 datasets for each parameter setting) under neutral evolution. As chromothripsis often occurs on one chromosome in a short time, we simulated for one or two cycles and biased DSBs towards chr1 (as in Fig. 1c). We only introduced interphase DSBs at the first cycle for some simulations with two cycles to compare with the effects of two consecutive cycles with interphase DSBs.

For parameter inference on simulated data, we set $S = 1$ with GRCh37/hg19 as a reference, since cancer genomes often underwent selection and genome sequencing data were often analysed by mapping to GRCh37/hg19. We did not model local fragmentation to reduce the number of parameters to estimate. To get realistic breakpoints, we sampled from breakpoints of a primary prostate cancer patient (ID: 2340225) in COSMIC[70], which has the most (14,626) SVs. We simulated until 10 cells, as it is computationally expensive to do ABC on a larger population size. We simulated 180 datasets with DSB rate per cycle $r \in \{10, 30\}$, fraction of unrepaired DSBs per cycle $f_u \in \{0.1, 0.3, 0.5\}$, and probability of WGD per cell $p_w \in \{0.1, 0.3, 0.5\}$, with 10 datasets for each parameter setting. To demonstrate the robustness of parameter inference, we also simulated data using breakpoints sampled from a primary breast cancer patient (ID: 2549320) in COSMIC, which included 951 SVs, with around 44% of the breakpoints overlapping with fragile sites. The simulation settings were identical to those for the prostate cancer patient, except that we used only the parameters shown in Fig. 5: $r = 30$, $f_u = 0.3$, and $p_w \in \{0.3, 0.5\}$.

We computed the overlap of simulated CNAs with cancer-related genes from COSMIC Cancer Mutation Census[70]. We used ShatterSeek[27] to detect chromothripsis from simulated SVs and CNAs. We counted the number of high-confidence chromothripsis events that have at least six interleaved intra-chromosomal SVs and at least seven segments with oscillating copy number states. Some of the simulated amplifications may be located in ecDNAs, which will cause higher-level amplifications and violation of the criteria of chromothripsis and lead to fewer detected chromothripsis events. We detected seismic amplification with the pipeline in ref. 8 and chromoplexy with Jabba[7]. Although most amplifications were located in ecDNAs, they can be integrated into the genome through circular recombination, which was not included here for simplicity. Hence, we treated ecDNAs as homogeneously staining regions that are reintegrated into the chromosome when detecting seismic amplification.

### Analysis of single-cell whole-genome sequencing data

Three mutational signatures were detected in the PDX datasets in ref. 43, where 8 datasets belong to HRD-Dup (homologous recombination deficiency-duplication) tumours which are enriched in BRCA1 mutations and small tandem duplications, 3 datasets belong to TD (tandem duplication) tumours which are enriched in large tandem duplications and *CDK12* mutations, and 12 datasets belong to FBI tumours that were used for our inference. As we assumed correct DNA replication other than chromosome fusion and no sequence homology when repairing DSBs to reduce complexity, our model is not suitable for SVs which were probably generated from HRD or other replication errors[25,39]. Therefore, we excluded 11 PDX datasets belonging to HRD-Dup and TD tumours and used the 20 remaining datasets for the subsequent analysis. We computed the empirical DSB rates for each dataset as the average number of subclonal SVs per clone expansion, namely the number of subclonal SVs divided by the number of clones minus 1.

For inference with ABC, we simulated data with GRCh37/hg19 as a reference under selection ($S = 1$) without local fragmentation. The fractions of breakpoints in telomere or centromere regions for all the datasets are less than 20% (Supplementary Data 2), so the exclusions of DSBs on these regions did not affect the inference. Because some datasets share clonal SVs and CNAs, which should be presented in the common ancestor of these clones, we started the simulation with the clonal SVs. To reduce stochasticity, the positions of new DSBs were sampled from the detected breakpoints based on their clonal prevalence, where breakpoints present in more clones were more likely to be sampled.

Note that the observed breakpoints or SVs are the results of both mutation and selection, which may suffer from false positives and false negatives due to the challenges in accurate detection[4]. In addition, not all the detected SVs arise from erroneous DNA repair and replication, such as SVs caused by transposable element insertion or virus integration. Therefore, the parameters inferred from real data only provided approximate estimates to better understand the underlying mutational processes.

### Analysis of bulk whole-genome sequencing data

To facilitate validation, we selected PCAWG samples with at least three clones and the presence of CNAs, SVs, ecDNAs, BFBs, and chromothripsis, including 148 samples without WGD and 5 samples with WGD. To examine more samples with WGD, we also included 27 additional samples that may not have complex SVs detected. We excluded samples with fewer than 10 SVs or an empirical DSB rate below five to ensure sufficient data for inference. Given the lack of subclonal haplotype-specific CNAs, we adjusted the computation of PGA to use allele-specific copy numbers and excluded pairwise divergence from the summary statistics. We excluded SVs without clonal assignments and computed breakpoint frequency distributions based on the

known probabilistic assignment of each SV to the detected clones[71]. Following a procedure similar to that for single-cell whole-genome sequencing data, we fit the bulk whole-genome sequencing data with ABC SMC.

The fraction of breakpoints in telomere or centromere regions is below 20% in all but four of the 111 datasets yielding ABC SMC results (Supplementary Data 3), suggesting that excluding DSBs in these regions has minimal impact on the inferences. We extracted mutational signatures from those computed in ref. 57 and copy number signature activities from[30]. To compute SV signatures, we applied SigProfiler[72] to 1815 PCAWG samples with SVs and then extracted the signatures for the 82 well-fit samples.

### Running ABC SMC on simulated data and whole-genome sequencing data

We used the function ABC SMC in ApproxBayes[64] to run the ABC SMC algorithm. In the algorithm, a set of sampled parameter values is propagated through a series of intermediate distributions (populations) with decreasing tolerances until the target tolerance $\epsilon$ is reached, where each sampled parameter value gets a weight based on its importance. We took 500 samples from each population to estimate the posterior distributions of parameters and set the maximum number of simulations to 1,000,000. To compute the posterior predictive distributions of summary statistics, we ran 500 simulations for each dataset using parameters sampled from the posterior distributions of the inferred parameters. We mainly varied the prior distributions of parameters and target tolerance and set the other parameters to the default values. For consistency, we assumed that the parameter priors were uniform for all the inferences on both simulated data and single-cell/bulk whole-genome sequencing data. The prior distributions of fraction of unrepaired DSBs and probability of WGD followed uniform distribution [0, 1]. For the inference on simulated data, we assumed the prior distribution of DSB rate followed uniform distribution [0, $M$], where the upper limit $M$ was 50, and set target tolerance $\epsilon$ to 0.2, which was sufficient to get accurate results in a reasonable amount of time (within 24 hours).

For the inference on single-cell and bulk whole-genome sequencing data, we assumed the prior distribution of DSB rate followed uniform distribution [0, $M$], where the upper limit $M$ was 50 for datasets with empirical DSB rate less than 50 and 100 otherwise. We set target tolerance $\epsilon$ to 0.5 for single-cell data and 0.2 for bulk data, because it would take a much longer time for convergence with a smaller $\epsilon$ (more than 72 hours). For 8 single-cell datasets (Fig. 6b) and 16 bulk datasets (Fig. 7) with wider posterior distributions, ABC SMC stopped at $\epsilon$ higher than 1 and 0.6 respectively, because the final $\epsilon$ was within 5% of the previous population. We define a dataset as poorly fit when its empirical DSB rate falls outside the 95% credible interval of the inferred DSB rate by a considerable margin, with absolute differences to the interval boundaries exceeding 10. Otherwise, the dataset is considered well-fit.

### Reporting summary

Further information on research design is available in the Nature Portfolio Reporting Summary linked to this article.

## Data availability

The simulated data and processed single-cell and bulk whole-genome sequencing data generated in this study have been deposited in Zenodo[73]. The initially processed single-cell whole-genome sequencing data used in this study are available at Zenodo (https://zenodo.org/record/6998936). The initially processed bulk whole-genome sequencing data used in this study are available at the International Cancer Genome Consortium Accelerating Research in Genomic Oncology (ICGC ARGO) data platform (https://platform.

icgc-argo.org/). The breakpoints used for simulating data during validation of parameter inference, as well as the cancer-related genes, were obtained from the Catalogue Of Somatic Mutations In Cancer (COSMIC) database (https://cancer.sanger.ac.uk/cosmic). The ecDNAs from the eccDNAdb database were obtained from Supplementary Table S2 of the corresponding paper[56]. Source data are provided with this paper.

## Code availability

The code for our computational model is freely available under MIT license at Github[74]. ShatterSeek v1.1, used for detecting chromothripsis, is available at https://github.com/parklab/ShatterSeek. Jabba v1.1, used for detecting chromoplexy, is available at https://github.com/mskilab-org/JaBbA. SeismicAmplification, used for detecting seismic amplification, is available at https://github.com/seismicon/SeismicAmplification. ApproxBayes v0.3.2, used for running ABC SMC, is available at https://github.com/marcjwilliams1/ApproxBayes.jl. Sig-Profiler v1.1.24, used for detecting structural variant signature, is available at https://github.com/AlexandrovLab/SigProfilerExtractor.

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

## Acknowledgements

C.B. and B.L. acknowledge funding from the Wellcome Trust (209409/Z/17/Z). The authors acknowledge the use of the UCL Myriad High Throughput Computing Facility (Myriad@UCL) and the UCL Department of Computer Science High Performance Computing Cluster, and associated support services, in the completion of this work. We thank Weini Huang for helpful discussion.

## Author contributions

C.B. conceived and supervised the study. B.L., S.W., and C.B. developed the model. S.W. implemented the initial version of the program. B.L. refined the program, conducted the analysis, and wrote the initial draft of the manuscript. W.C. provided critical interpretation of the results. All authors reviewed, revised, and approved the final manuscript.

## Competing interests

The authors declare no competing interests.
