## [Transparent Peer Review file · Nature Communications]

Cell-cycle dependent DNA repair and replication unifies patterns of chromosome instability

Corresponding Author: Professor Chris Barnes

Version 0:

Reviewer comments:

Reviewer #1

(Remarks to the Author)

Lu et al. present a novel computational model for simulating structural variant (SV) generation. The analyses presented in the manuscript, while comprehensive and original in developing a model for SV generation in cancer genomes, could be considered somewhat preliminary.

The model's predictions and simulations are primarily validated using comparisons with known patterns of SVs in cancer, which appear to be largely of a qualitative nature (or at least the text does not make clear that otherwise is the case). For example, that the simulation can generate complex SV patterns apparently resembling chromothripsis and BFB cycling-resulting patterns is interesting. However, a more quantitative correlation of these simulation results with real-life observations of these phenomena (e.g. frequencies across patient-derived samples or cell lines, distributions of inter-breakpoint distances etc) would be beneficial for robust validation. Currently the validation appears somewhat limited.

The model makes substantial simplifying assumptions. Prominently, it ignores heterogeneity in risk of breakpoint locations, and disregards sequence homology in DSB repair (both of these are transparently discussed – kudos for that). However, there are other assumptions implicit e.g. it is not clear their adjacencies consider spatial arrangement of chromatin (the discussion suggests they ignore it), and also it was not clear whether model accounts for that selection acts on genes with different strengths (e.g. I am not sure that EGFR and CARD11 would be generating a similar selective advantage if amplified). While potentially admissible in a preliminary study, these simplifications might limit the model's ability to clarify mechanistic steps in SV formation in real cancer genomes (e.g. the interesting proposal that chromothripsis accumulates over >1 cell cycle), and this appears to not have been tested; we do not know if these assumptions are safe to make. At the very least they could have included the e.g. known fragile sites to test how robust the model's inferences are to changes in these assumptions and/or to key parameters.

There are worries about ability to generalize and applicability to different biological contexts. The study primarily focuses on the development and initial validation of the computational model. While the results appear promising in that they recapitulate some prominent types of SV patterns seen across cancer genomes, what is currently less convincing is whether the parameter space exploration (DSB rate per cycle, percentage of unrepaired DSBs, scale of local fragmentation...) could have yielded this level of agreement at random chance. This is hard to judge intuitively and thus merits a rigorous text. Also, a quantitation of the agreement to actual data would be needed to show the applicability of the model to a wide range of cancer types (with variable repertoires of driver genes under selection) and maybe more importantly genomic contexts (e.g. activity of mutational processes) would require further investigation.

Overall, while this study is conceptually interesting and timely, I also think it is not quite ready, and addressing these aspects mentioned above would strengthen it such that it represents a clearer contribution to the field of SV mutagenesis and cancer evolution.

In addition, the manuscript text could also be much better written to improve readability (avoid jargon), reduce clutter (resulting e.g. from mixing results from the current study and previous observations) and perhaps most importantly vague statements where the support in the data is unclear should be removed to focus the text. As an example “experimental studies have recently been done to investigate whether co-localization of multiple DSBs can stimulate chained inter-chromosomal and intra-chromosomal translocations typical for chromoplexy [45]. This is now confirmed in our simulations

which may help to further understand the processes generating chromoplexy by fitting experimental data in the future.” it was not clear what in their data supported this statement and what was the rigorous test to show what degree of DSB co-localization yields the chromoplexy-like pattern and what is the statistical support for this resemblance. For a certain claim, if specific data points cannot be cited to support it, it is probably best removed, streamlining the text to make it more focused on the better supported parts of the study.

(Remarks on code availability)

Reviewer #2

(Remarks to the Author)

The authors developed a computational model of the cell cycle to understand how structural variations (SVs) form when cells repair double-strand breaks. This model reveals the relationship between different processes, such as the BFB cycle, chromothripsis, and ecDNA.

Moreover, they utilized Bayesian inference to infer various statistics affecting SV formation from cell sequencing data. To validate their model, they conducted tests using both simulated data and real single-cell whole-genome sequencing data.

Overall this is a clear paper with a strong and timely contribution.

Here are some minor questions to clarify and suggestions to make the paper even stronger:

1. As previously demonstrated and emphasized in this study, chromothripsis emerges as a significant driver of extrachromosomal DNAs (ecDNAs), prompting further exploration within simulated data. However, despite the analysis of both chromothripsis and ecDNAs in Fig. 3, the relationship between ecDNA and chromothripsis remains unclear. In Table S3, both ecDNA and chromothripsis are observed in cycle 4 under the neutral model, while in the selection model, they appear in cycles 3 and 5, respectively. It is intriguing to examine the lineage following a chromothripsis event and assess whether the rate of ecDNA appearance is influenced by chromothripsis. Please verify if this observation holds true through repeated simulations.
2. I'm curious about the WGD occurrence in the simulations discussed in the section "The simple repair process model explains the formation of complex SVs" and "Role of cell cycle in formation of complex SVs including chromothripsis." Exploring the interaction between WGD, ecDNA, and chromothripsis in the simulations could provide valuable insights, especially given the correlation observed between ecDNAs and WGD in Fig. 6d. It might be helpful to also show the WGD numbers in Table S3. In A9, it is shown that there is no correlation between the inferred probability of WGD per cell and the number of ecDNAs, but how about the real occurrence of WGD?
3. How many misrepaired double-strand breaks (DSBs) were intentionally introduced to observe chromoplexy in Figure 2f? Was this test repeated to assess the frequency of chromoplexy occurrence?
4. In Fig 3, we observed a decrease in the number of chromothripsis events on chr1, transitioning from the initial state of [2 cycles (interphase DSB at 1st cycle)] to [2 cycles] across various settings (with 3 unrepaired DSBs per cycle and 2 DSP rates per cells) when 10 DSBs occurred in local fragmentation. This decrease was attributed to the stringent criteria employed in event detection. Could you please elaborate on the specific criteria that might elucidate this decline?
5. In Fig. 3e, given the focus of the analysis on inducing double-strand breaks (DSBs) on chr1 and the subsequent examination of cells with chromothripsis on chr1 in Fig. 3c, please also include the number of cells with seismic amplifications on chr1.
6. In Fig. 4b, the fraction of cells with ecDNA exceeds 94% in all cases. Is this level of abundance realistic?
7. Fig 4b: Please also show the changes of the number of ecDNAs with different copy numbers for 1000, 3000 and 5000 cells.
8. Fig A5 and 4b: It would be interesting to see the relation of the selection coefficient and the mean copy number of ecDNA in the cells.
9. Figure 5f: The error that is reported for mean ecDNAs per cell is quite high and the fact that it became more accurate when there were more ecDNAs in the data could be a sign that the tolerance threshold should be decreased as the range of values also decreased. In the related section (Running ABC SMC on simulated data and single-cell whole-genome sequencing data) I could not find the tolerance value and the prior distribution for the number of ecDNA per cell and the number of fusions per cycle. A plot of the evolution of the tolerance threshold (ϵ) over iterations or acceptance rate could be informative as well.
10. Fig. A1 needs legend and the explanation of the arrows.
11. It is helpful to see the list of all parameters and default values in Table S1. But I assume that these default values were

changed for the different simulations across the paper. If that is true, please extend the table to include the actual values chosen for each simulation.

12. Equation 1 has very specific values for d and c . How were they chosen?

(Remarks on code availability)

Reviewer #3

(Remarks to the Author)

(Remarks on code availability)

Reviewer #4

(Remarks to the Author)

The manuscript 'Cell-cycle dependent DNA repair and replication unifies patterns of chromosome instability' by Barnes and colleagues presents a computational cell-cycle model to generate structural variations from end-joining repair and replication after double strand break formation and infer the parameters for modeling generation with Bayesian inference. The model provides quantitative information on the relationship between different classes of complex structural variations including BFB cycle, chromothripsis, seismic amplification, and eccDNA.

I found the modeling approach to integrate DSB, complex SV generation, and branching process under neutrality and selection to be novel and of potential interest. Unlike the point mutations, evolutionary dynamics of complex SV events under selection remains poorly established. Since many oncogenic alterations arise via complex SVs mentioned here, this is an important area. However, I have several major questions about the key assumptions, simulation settings, and validation.

First, in the model the DSBs are generated randomly throughout the genome in the G1 phase. In somatic cells, however, specific mechanisms such as fork collapse within the replication factory results in DNA breaks, which may violate the above assumption. There are more long range chromosomal interactions among the break ends than that expected by random chance. On a similar note, it may violate the assumption that pairing of DSBs is random, or merely linear distance-dependent along a chromosome. Ultimately, these assumptions can bias the context and size distribution of SVs expected under neutral evolution and selection (see PMID: 17137790; PMID: 21962511).

Second, the model for chromothripsis assumes that DSBs are generated until $nd+1$ cycles but none thereafter. It may be a convenient assumption to study evolutionary patterns of a chromothripsis event, but the assumption on subsequent genomic stability of the affected region may not be realistic. Rearranged genomic segments are likely to alter chromatin folding and replication timing contributing to persistent instability and heterogeneity at the affected locus. (example: Fig-8 of Voronina et al. Nature Comm, 2020).

Third, many oncogenes and tumor suppressor genes function in a tissue-dependent manner, such that the density of tumor suppressor and oncogenes to assess the clonal fitness may be simplistic. Chromosomal ploidy and local copy numbers may affect the pointy mutation rate in a cancer gene in the affected region, that has potential to ultimately changing the fitness landscape (e.g. copy number of EGFR and T790M mutation therein; also see Selmecki et al. Nature, 2015).

Fourth, ecDNA copy number increase can confer fitness advantage, but at some point the high copy numbers result in fitness cost to the cell for maintaining the ecDNA (Verhaak et al. Nature Rev Genet, 2020, and references 51-52 therein). Likewise, unrepaired DSBs also result in a fitness cost to the cell. Such issues were not suitably modeled during simulation in this paper.

Fifth, validation analyses could be more rigorous, ideally using data from barcoded cell populations from in vitro experiments, or deeply profiled human tumors (with high mutation burden) to secure deep clonal phylogeny. Fig-6 presented a valuable analysis, but I felt that the pseudobulk analysis, as presented, under-utilized the potential of the dataset. However, I was not convinced by the interpretation that dismissed unfavorable results as - 'For a few datasets, the inferred DSB rates were more different from empirical DSB rates (Fig. A7a), which could be due to SVs generated from mechanisms not captured in our model such as HRD or replication errors'. Again, these issues call for more validation analyses from different tumor types.

(Remarks on code availability)

Version 1:

Reviewer comments:

Reviewer #1

(Remarks to the Author)

Overall, the authors made substantial improvements in addressing my technical and validation concerns. Regarding quantitative validation, they added correlations with real PCAWG data, provided comparisons of inter-breakpoint distances between simulated and real data. Regarding parameter space exploration, they have also strengthened this aspect by demonstrating robustness through repeated simulations, and calculating fraction of well-fit summary statistics. I do also appreciate the transparency in discussing the model limitations regarding random breakpoint locations vs fragile sites, DNA repair simplifications and impact of chromatin organization.

Areas that could be further improved (in future work) involve mechanistic testing of different processes that contribute to SV formation, more developed selection models that incorporate tissue specific effects and complex fitness landscapes (epistasis), and finally the lack of incorporation of chromatin organization remains a significant simplification. All this seems reasonable given the scope of the study (which already at this stage presents a sufficient step forward), and perhaps also given the computational tractability requirements.

The revised manuscript represents a meaningful advance in modeling SV evolution in cancer, likely to be valuable and interesting to an audience of cancer biologists, evolutionary biologists and the DNA repair field.

(Remarks on code availability)

Reviewer #2

(Remarks to the Author)

In general, very happy with authors' response. Some minor issues remain:

In both the responses to the reviews and the paper, it has been repeatedly mentioned that time constraints have affected the ability to optimize the tolerance parameter, thereby impacting result accuracy. Please provide an estimated runtime for the inference process to give a clearer understanding of the time requirements.

In Figure 7, the subfigures are misaligned and lack proper labelling. Additionally, the colours used are not easily distinguishable, making it difficult to interpret the data effectively.

Figure 7: How are you calculating the number of ecDNA per cell in the bulk data? What assumptions or estimates did you use to determine the number of cells?

Figure 8:

report the correlation and p-value for all the subplots.

What criteria are used to define well-fit samples in both single-cell and bulk datasets?

Comparing the number of ecDNA in single-cell and bulk datasets in section "Fitting the model to bulk whole-genome sequencing data of multiple cancer types" does not seem feasible. This is because the number of ecDNA elements is highly dependent on the type of sample being analysed. It is essential to specify the types of samples used in Figure 8 for both single-cell and bulk datasets. For example, a higher percentage of ecDNA is expected in late-stage oesophageal adenocarcinoma compared to early-stage cases (DOI: 10.1038/s41586-023-05937-5).

With simulated data, it is shown that the number of ecDNAs per cell is significantly higher in cells with WGD compared to those without WGD. Similarly, in the real data, the section "Fitting the model to bulk whole-genome sequencing data of multiple cancer types" suggests that the posterior mean number of ecDNA significantly increases with the inferred probability of WGD. However, in Figure 7, the samples with WGD appear to have a lower inferred mean number of ecDNAs. Could you clarify this?

Please clearly define the specific patterns you considered as chromothripsis-like patterns and elaborate on what you mean by chromothripsis-related events in the results section to enhance clarity.

(Remarks on code availability)

Reviewer #3

(Remarks to the Author)

(Remarks on code availability)

Reviewer #4

(Remarks to the Author)

I have no further queries on the revised manuscript.

(Remarks on code availability)

REVIEWER COMMENTS

Reviewer #1 (Remarks to the Author):

Lu et al. present a novel computational model for simulating structural variant (SV) generation. The analyses presented in the manuscript, while comprehensive and original in developing a model for SV generation in cancer genomes, could be considered somewhat preliminary.

The model's predictions and simulations are primarily validated using comparisons with known patterns of SVs in cancer, which appear to be largely of a qualitative nature (or at least the text does not make clear that otherwise is the case). For example, that the simulation can generate complex SV patterns apparently resembling chromothripsis and BFB cycling-resulting patterns is interesting. However, a more quantitative correlation of these simulation results with real-life observations of these phenomena (e.g. frequencies across patient-derived samples or cell lines, distributions of inter-breakpoint distances etc) would be beneficial for robust validation. Currently the validation appears somewhat limited.

1.1 Response:

As our model was initially designed for single-cell whole-genome sequencing data where it is hard to detect SVs, we did not correlate our simulations to real observations of complex SVs. However, we have now enhanced the simulation program to allow inferences on bulk whole-genome sequencing data from the Pan-Cancer Analysis of Whole Genomes (PCAWG) Consortium and added the analysis as Fig. 7 in the main manuscript.

PCAWG datasets have been extensively studied, revealing multiple types of complex SVs including chromothripsis, BFBs, and ecDNAs. We compared the posterior predictive distributions of the numbers of BFBs (chromosome fusions) and ecDNAs with the observed distributions. We also computed the overlap of inferred ecDNAs with ecDNAs collected from patient tissues, patient-derived xenografts, and cell lines from the database eccDNAdb (DOI: 10.1038/s41388-022-02286-x). The results show that the number of inferred ecDNAs generally increased with the number of SV-overlapping ecDNAs in the database. The known number of chromothripsis events also significantly correlated with the posterior means of ecDNAs. However, the known numbers of events, especially ecDNAs and BFBs, are likely underestimations, suggesting that prior analyses may have missed some complexities. We have added these results as Fig. 8 in the main manuscript.

To get more realistic breakpoints, we sampled breakpoints from real data during Bayesian inference. To assess the consistency of inter-breakpoint distances between simulated and real data, we simulated 500 datasets for each well-fit single-cell and PCAWG dataset using parameters sampled from the posterior distributions of inferred parameters. We then compared the median of generated posterior predictive distributions with the real data for each dataset using quantile-quantile (QQ) plots and Kolmogorov-Smirnov (KS) tests. The simulated distances were mostly consistent or slightly smaller than the observed values in 19 single-cell datasets, among which 13 datasets have very similar simulated and observed distributions, as indicated by p-values from KS tests below 0.05 (Supplementary Fig. 21). For 82 PCAWG datasets, the simulated distances were mostly consistent or slightly larger than the observed distances, among which 28 datasets have very similar simulated and observed distributions (Supplementary Fig. 27).

The model makes substantial simplifying assumptions. Prominently, it ignores heterogeneity in risk of breakpoint locations, and disregards sequence homology in DSB repair (both of these are transparently discussed – kudos for that). However, there are other assumptions implicit e.g. it is not

clear their adjacencies consider spatial arrangement of chromatin (the discussion suggests they ignore it), and also it was not clear whether model accounts for that selection acts on genes with different strengths (e.g. I am not sure that EGFR and CARD11 would be generating a similar selective advantage if amplified). While potentially admissible in a preliminary study, these simplifications might limit the model's ability to clarify mechanistic steps in SV formation in real cancer genomes (e.g. the interesting proposal that chromothripsis accumulates over >1 cell cycle), and this appears to not have been tested; we do not know if these assumptions are safe to make. At the very least they could have included the e.g. known fragile sites to test how robust the model's inferences are to changes in these assumptions and/or to key parameters.

1.2 Response:

We did not explicitly consider the spatial arrangement of chromatin in our current model to avoid additional parameters which are hard to estimate from whole-genome sequencing data. However, we mentioned incorporating 3D nuclear distances to derive contact probabilities as future work in Discussion, which implicitly includes the effect of chromatin organization. To clarify the point, we have now added more details about the distances in Fig. 1 Caption and Discussion in the main manuscript.

Our current model of selection considers both the density and potency of tumour suppressor genes (TSGs) and oncogenic genes (OGs) on a particular arm or chromosome, where the potency is measured by their rank on the respective lists obtained from the TUSON Explorer. TUSON Explorer ranks genes by an overall significance derived from their mutational profile. The Charm/Chrom (TSG-OG) score quantifies the positive or negative impact on growth and survival that wild-type OGs or TSGs typically provide to a chromosomal arm or whole chromosome. The score is higher for arms or chromosomes with a greater number or potency of TSGs and is lower for those with a higher number or potency of OGs. Therefore, selection strengths of various genes, such as EGFR (ranked 10th in the OG list) and CARD11 (ranked after 4000 in both OG and TSG lists), are considered in our model. We have clarified our model of selection with more details in Methods.

Although our simulations suggest that chromothripsis is more likely formed in two cell cycles, chromothripsis was also formed in the simulations when there was more local fragmentation in one cell cycle. This is consistent with experimental studies showing that chromothripsis occurred infrequently during the first interphase and became more prevalent during the second cell cycle (DOI: 10.1126/science.aba0712). We have rephrased the text in the main manuscript to avoid confusion.

We also examined the overlap between random breakpoints and fragile sites in the simulated "true" data for the inferences presented in Fig. 5. The results show that around 40% of the breakpoints overlap with fragile sites (Supplementary Fig. 35a). This indicates that a substantial fraction of DSBs occur at fragile sites in our simulations, although DSBs were introduced randomly. We will investigate the effect of DSBs explicitly biased towards fragile sites in future work.

When fitting the model to data (Fig. 5-7), we sampled from known breakpoints to reduce the search space. For example, when fitting the model to simulated data (Fig. 5), we showed that accurate inferences could be obtained by randomly sampling from breakpoints of a primary prostate cancer sample with the most (14,626) SVs in COSMIC. To show the robustness of our model, we did Bayesian inferences with breakpoints from a primary breast cancer sample with 951 SVs and around 44% of the breakpoints overlapping with fragile sites. Using the same parameters as those in Fig. 5b-e, the results of Bayesian inferences were consistent with those obtained using the breakpoints from the prostate cancer sample (Supplementary Fig. 17). Moreover, the simulated data have around 30% of the breakpoints overlapping with fragile sites (Supplementary Fig. 35b).

There are worries about ability to generalize and applicability to different biological contexts. The study primarily focuses on the development and initial validation of the computational model. While the results appear promising in that they recapitulate some prominent types of SV patterns seen across cancer genomes, what is currently less convincing is whether the parameter space exploration (DSB rate per cycle, percentage of unrepaired DSBs, scale of local fragmentation...) could have yielded this level of agreement at random chance. This is hard to judge intuitively and thus merits a rigorous text. Also, a quantitation of the agreement to actual data would be needed to show the applicability of the model to a wide range of cancer types (with variable repertoires of driver genes under selection) and maybe more importantly genomic contexts (e.g. activity of mutational processes) would require further investigation.

1.3 Response:

To show our exploration of parameter space is robust to random noise, we added statistical test results to measure the significance of differences under different parameter settings in Fig. 3 and A4, which show how the number of events varies with parameter changes in simulations. We also improved the related text in the main manuscript.

To validate parameter inference on simulated data, we repeated the simulations with the same parameters ten times to show the agreement is not due to random chance. The accuracies of the posterior means of inferred parameters for six and all of the parameter settings are shown in Fig. 5b-f and Extended Data Fig. 6, respectively. Supplementary Fig. 14 and 15 show the posterior distributions of inferred parameters for each run, where more than 90% of real values fall within two standard deviations of the distributions for all five parameters (Supplementary Table 6). As we used uniform distributions as priors for the inferences, the concentrated posterior distributions across runs clearly indicate the likelihood that parameters were driven by data rather than chance alone. Supplementary Fig. 16 shows the posterior predictive distributions of summary statistics used in inference from one simulated dataset, where all the observed values fell within two standard deviations of the distributions, suggesting that the inferences fit the data well. We calculated the fraction of summary statistics whose observed values fell within two standard deviations of the corresponding posterior predictive distributions, denoted as P_w . The values of P_w across all simulated datasets suggest that more than 74% of datasets have $P_w = 1$ and only four datasets have $P_w < 0.9$, with the minimal value of P_w being 0.83.

When fitting single-cell whole-genome sequencing data, we show the agreement of inferences to actual data where we compared estimated DSB and WGD rates with empirical approximations in Extended Data Fig. 7. Supplementary Fig. 20 further shows the posterior predictive distributions of summary statistics used in inference from one single-cell dataset, where observed values for 15 out of 24 summary statistics fell within two standard deviations of the distributions. The values of P_w across all single-cell datasets suggest that most datasets have $P_w > 0.5$ except four datasets.

To emphasize the agreement of our simulations and inferences to reality, we have added these additional details in the main manuscript.

To show the generalizability and applicability of our model to a wide range of biological contexts, we fit our model to 111 bulk whole-genome sequencing datasets from PCAWG and got good results for 82 samples across ten cancer types (Fig. 7,8, Supplementary Table 8). The mutational, copy number, and SV signatures suggested variable genomic context or different activities of different mutational processes in each sample (Supplementary Fig. 32-34, Supplementary Table 9). We added the results

as a new section “Fitting the model to bulk whole-genome sequencing data of multiple cancer types” in the main manuscript.

Overall, while this study is conceptually interesting and timely, I also think it is not quite ready, and addressing these aspects mentioned above would strengthen it such that it represents a clearer contribution to the field of SV mutagenesis and cancer evolution.

In addition, the manuscript text could also be much better written to improve readability (avoid jargon), reduce clutter (resulting e.g. from mixing results from the current study and previous observations) and perhaps most importantly vague statements where the support in the data is unclear should be removed to focus the text. As an example “experimental studies have recently been done to investigate whether co-localization of multiple DSBs can stimulate chained inter-chromosomal and intra-chromosomal translocations typical for chromoplexy [45]. This is now confirmed in our simulations which may help to further understand the processes generating chromoplexy by fitting experimental data in the future.” it was not clear what in their data supported this statement and what was the rigorous test to show what degree of DSB co-localization yields the chromoplexy-like pattern and what is the statistical support for this resemblance. For a certain claim, if specific data points cannot be cited to support it, it is probably best removed, streamlining the text to make it more focused on the better supported parts of the study.

1.4 Response:

We have explained the jargon introduced in the Introduction of the main manuscript. We have also eliminated word clutters by separating the descriptions of our results and previous observations.

We agree that specific data are needed to support the statement on how the experimental studies are confirmed in our simulations of chromoplexy. The main point we want to emphasize is that our model can simulate the co-localization of multiple DSBs and associated chromoplexy patterns. While the experimental studies are still ongoing, our model may be used to fit the generated data in the future to better understand the processes. To further assess the frequencies of chromoplexy, we performed 10 repeated tests with different random seeds. Compared to the results in Fig. 2f and A3, repeated simulations indicate that more cells are likely to have chromoplexy under the same parameter settings (Supplementary Fig. 6). To determine the minimum number of misrepaired DSBs required to generate chromoplexy, we also conducted 10 repeated simulations with progressively fewer misrepaired DSBs per cycle (5, 2, 1) on random chromosomes in each cell cycle. The results suggest that chromoplexy started to appear when there was more than one DSB per cycle (Supplementary Fig. 6). We have incorporated these results and improved the vague statement about chromoplexy in the main manuscript.

Reviewer #2 (Remarks to the Author):

The authors developed a computational model of the cell cycle to understand how structural variations (SVs) form when cells repair double-strand breaks. This model reveals the relationship between different processes, such as the BFB cycle, chromothripsis, and ecDNA.

Moreover, they utilized Bayesian inference to infer various statistics affecting SV formation from cell sequencing data. To validate their model, they conducted tests using both simulated data and real single-cell whole-genome sequencing data.

Overall this is a clear paper with a strong and timely contribution.

Here are some minor questions to clarify and suggestions to make the paper even stronger:

1. As previously demonstrated and emphasized in this study, chromothripsis emerges as a significant driver of extrachromosomal DNAs (ecDNAs), prompting further exploration within simulated data. However, despite the analysis of both chromothripsis and ecDNAs in Fig. 3, the relationship between ecDNA and chromothripsis remains unclear. In Table S3, both ecDNA and chromothripsis are observed in cycle 4 under the neutral model, while in the selection model, they appear in cycles 3 and 5, respectively. It is intriguing to examine the lineage following a chromothripsis event and assess whether the rate of ecDNA appearance is influenced by chromothripsis. Please verify if this observation holds true through repeated simulations.

2.1 Response:

To better understand the relationship between ecDNAs and chromothripsis, we examined the cell lineages in Extended Data Fig. 1 through repeated simulations with the same random seed until different population sizes (Supplementary Fig. 1-2). The first chromothripsis-like pattern, along with 8 ecDNAs, is detected in cell 12 at the 5th cell cycle when there are seven cells in the final population. This is caused by local fragmentation in cell 9, the grandparent of cell 12.

In Table S3 (Supplementary Table 3), we previously reported “cycle appearing” as the earliest cell cycle of all the cells carrying ecDNAs. We realized that this terminology is misleading, because it does not account for population size, which reflects the time point when the cells divide. For example, a cell carrying ecDNAs might appear after the fifth cell division at an earlier time point when there is a smaller population size of seven cells, whereas another cell carrying ecDNAs might appear after the third cell division at a later time point when there is a slightly larger population size of ten cells (Supplementary Fig. 1). To eliminate confusion, we have changed “cycle appearing” to “time appearing” which is measured by population size.

In general, the cell lineage trees and the observed numbers and timing of complex SVs over time suggest that: 1) chromothripsis and ecDNAs tend to occur concurrently following local fragmentation; 2) all cells exhibiting chromothripsis-like patterns contain ecDNAs and originate from ancestral cells that underwent local fragmentation; 3) ancestors of cells lacking ecDNAs show little to no local fragmentation.

We repeated the simulation shown in Extended Data Fig. 1 50 times, each with a different random seed. The results show that cells with both chromothripsis and ecDNAs have slightly higher numbers of ecDNAs (Supplementary Fig. 3a). Because the chromothripsis-like patterns detected by ShatterSeek may have false negatives, we also consider known local fragmentations as general chromothripsis events, which represent mutational processes in which large stretches of a chromosome undergo massive rearrangements in a single, catastrophic event. The results suggest that the numbers of ecDNAs and chromothripsis-like patterns detected by ShatterSeek are significantly higher after local fragmentation (Supplementary Fig. 3b). In summary, these results suggest that the emergence of ecDNA is influenced by chromothripsis-related events, particularly local fragmentation. We have incorporated these results in the main manuscript.

2. I'm curious about the WGD occurrence in the simulations discussed in the section "The simple repair process model explains the formation of complex SVs" and "Role of cell cycle in formation of complex SVs including chromothripsis." Exploring the interaction between WGD, ecDNA, and chromothripsis in the simulations could provide valuable insights, especially given the correlation observed between ecDNAs and WGD in Fig. 6d. It might be helpful to also show the WGD numbers in Table S3. In A9, it is shown that there is no correlation between the inferred probability of WGD per

cell and the number of ecDNAs, but how about the real occurrence of WGD?

2.2 Response:

The simulations discussed in the section "The simple repair process model explains the formation of complex SVs" (Fig. 2) and "Role of cell cycle in formation of complex SVs including chromothripsis" (Fig. 3) do not have WGDs because we wanted to prevent WGD from affecting other SVs and hence set the probability of WGD to 0.

We performed additional simulations with increasing probabilities of WGD while keeping all other parameters the same as those used for the simulation with local fragmentation in Extended Data Fig. 1. All the cells underwent WGD when the probability of WGD exceeded 0.123. The numbers of cells with WGD for different probabilities are listed in Table S3 (Supplementary Table 3): 0.1 (34 out of 200 cells), 0.122 (126 out of 200 cells), and 0.124 (all cells). The observed frequency and timing of complex SVs suggest that ecDNAs and chromothripsis were often generated simultaneously, followed closely by seismic amplifications. We repeated the simulations 50 times when the probability of WGD was 0.1 and 0.122, respectively. The results suggest that cells with WGD have significantly more ecDNAs and chromothripsis per cell than cells without WGD and their mean numbers of ecDNAs and chromothripsis per cell are often significantly higher as well (Supplementary Fig. 4-5). We have incorporated these results in the main manuscript.

We added a plot showing the correlations between the real parameters used in simulating "true" data for the inferences presented in Fig. 5 (Supplementary Fig. 23). There is a slight yet significantly positive correlation between the probability of WGD and the number of chromosome fusions. This correlation was not detected in the inferred parameters due to the slight underestimation of chromosome fusions (Fig. 5e). There is no significant correlation between the probability of WGD and the number of ecDNAs, although the numbers of ecDNAs per cell are significantly higher in cells with WGD than those in cells without WGD (Supplementary Fig. 24).

3. How many misrepaired double-strand breaks (DSBs) were intentionally introduced to observe chromoplexy in Figure 2f? Was this test repeated to assess the frequency of chromoplexy occurrence?

2.3 Response:

We introduced 10 misrepaired DSBs per cycle on random chromosomes in each cell cycle until 5 or 10 cells to generate chromoplexy (Methods). To avoid confusion, we have moved the exact values to Results. To further assess the frequencies of chromoplexy, we performed 10 repeated tests with different random seeds. Compared to the results in Fig. 2f and A3, repeated simulations indicate that more cells are likely to have chromoplexy under the same parameter settings (Supplementary Fig. 6). To determine the minimum number of misrepaired DSBs required to generate chromoplexy, we also conducted 10 repeated simulations with progressively fewer misrepaired DSBs per cycle (5, 2, 1) on random chromosomes in each cell cycle. The results suggest that chromoplexy started to appear when there was more than one DSB per cycle (Supplementary Fig. 6). We have incorporated these results in the main manuscript.

4. In Fig 3, we observed a decrease in the number of chromothripsis events on chr1, transitioning from the initial state of [2 cycles (interphase DSB at 1st cycle)] to [2 cycles] across various settings (with 3 unrepaired DSBs per cycle and 2 DSP rates per cells) when 10 DSBs occurred in local fragmentation. This decrease was attributed to the stringent criteria employed in event detection.

Could you please elaborate on the specific criteria that might elucidate this decline?

2.4 Response:

The criteria employed in detecting high-confidence chromothripsis events by ShatterSeek include: 1) at least seven adjacent segments oscillating between two copy number states, 2) at least six interleaved intra-chromosomal SVs, 3) fragment joins test, and 4) either the chromosomal enrichment test or the exponential distribution of breakpoints test.

To illustrate how the number of detected chromothripsis events decreases with the application of successive criteria, we plotted the number of cells exhibiting chromothripsis events on chr1 after imposing each criterion. Additionally, we included two less stringent criteria on the number of adjacent segments oscillating between copy number states. The results suggest that the first two criteria play a critical role in limiting the number of chromothripsis events detected on chr1 (Supplementary Fig. 8).

We have added the specific criteria in the main manuscript to avoid confusion.

5. In Fig. 3e, given the focus of the analysis on inducing double-strand breaks (DSBs) on chr1 and the subsequent examination of cells with chromothripsis on chr1 in Fig. 3c, please also include the number of cells with seismic amplifications on chr1.

2.5 Response:

Because most seismic amplifications occurred on chr1 which has the highest number of breakpoints, the number of cells with seismic amplifications is only slightly larger than the count on chr1, except for four cases. We now show the number of cells with seismic amplifications on chr1 in Supplementary Fig. 7.

6. In Fig. 4b, the fraction of cells with ecDNA exceeds 94% in all cases. Is this level of abundance realistic?

2.6 Response:

When the fraction of ecDNAs in the simulated data is as high as 94%, these ecDNAs can be seen as clonal. The clonality of ecDNAs is still understudied with limited data (DOI: 10.1038/s41576-022-00521-5). The theoretical predictions suggest that the fraction of cells with an ecDNA under positive selection will approach 1 in large populations (DOI: 10.1038/s41588-022-01177-x). Additionally, clonal ecDNAs have been reported in biopsies of two patients with Barrett's oesophagus (DOI: 10.1038/s41586-023-05937-5). Therefore, the high level of ecDNA abundance in our simulations seems realistic.

7. Fig 4b: Please also show the changes of the number of ecDNAs with different copy numbers for 1000, 3000 and 5000 cells.

2.7 Response:

We added a plot to show the changes of the number of ecDNAs with different copy numbers for 1000, 3000 and 5000 cells, which exhibit similar distributions, with the maximum copy number increasing over time (Supplementary Fig. 10).

8. Fig A5 and 4b: It would be interesting to see the relation of the selection coefficient and the mean copy number of ecDNA in the cells.

2.8 Response:

We added a plot to show the correlations between selection coefficients and mean copy numbers of ecDNA in the cell, which reveals a small yet significant positive correlation with the selection coefficient as cell population size increases, suggesting a tendency for cells with higher ecDNA copy numbers to be favored by positive selection (Supplementary Fig. 11).

9. Figure 5f: The error that is reported for mean ecDNAs per cell is quite high and the fact that it became more accurate when there were more ecDNAs in the data could be a sign that the tolerance threshold should be decreased as the range of values also decreased. In the related section (Running ABC SMC on simulated data and single-cell whole-genome sequencing data) I could not find the tolerance value and the prior distribution for the number of ecDNA per cell and the number of fusions per cycle. A plot of the evolution of the tolerance threshold (ϵ) over iterations or acceptance rate could be informative as well.

2.9 Response:

We specified the setting of the target tolerance ϵ later in the section, which was 0.2 for simulated data and 0.5 for single-cell whole-genome sequencing data. To avoid confusion, we have added “target tolerance” before “ ϵ ” in Methods. We have plotted the tolerance values over iterations and the acceptance rates for inferences on both simulated data (Supplementary Fig. 12-13) and whole-genome sequencing data (Supplementary Fig. 18-19, 25-26). For simulated datasets containing more ecDNAs, target tolerances were achieved in fewer iterations and with higher acceptance rates. Reducing the target tolerance would yield more accurate results but would also increase the computational time required. For consistency and convenience, we used the same target tolerance across all the datasets.

There are no prior distributions for the number of ecDNA per cell and the number of fusions per cycle. These two metrics are summary statistics derived from the simulations, which are often not directly measurable through genome sequencing data. Instead, they are inferred from the posterior predictive distributions of ABC SMC. This is indicated in the latter part of the first paragraph of Section “Validation of parameter inference on simulated data”. We have also added additional details on computing the posterior predictive distributions of summary statistics in Methods.

10. Fig. A1 needs legend and the explanation of the arrows.

2.10 Response:

We have added the legend for copy number heatmaps and explanations of the arrows.

11. It is helpful to see the list of all parameters and default values in Table S1. But I assume that these default values were changed for the different simulations across the paper. If that is true, please extend the table to include the actual values chosen for each simulation.

2.11 Response:

We have added another column in Table S1 (Supplementary Table 1) to show the values used for the different simulations.

12. Equation 1 has very specific values for d and c. How were they chosen?

2.12 Response:

The values for d and c are chosen based on previous studies. We have added the explanations in Methods for clarity.

Reviewer #3 (Remarks to the Author):

Reviewer #4 (Remarks to the Author):

The manuscript 'Cell-cycle dependent DNA repair and replication unifies patterns of chromosome instability' by Barnes and colleagues presents a computational cell-cycle model to generate structural variations from end-joining repair and replication after double strand break formation and infer the parameters for modeling generation with Bayesian inference. The model provides quantitative information on the relationship between different classes of complex structural variations including BFB cycle, chromothripsis, seismic amplification, and eccDNA.

I found the modeling approach to integrate DSB, complex SV generation, and branching process under neutrality and selection to be novel and of potential interest. Unlike the point mutations, evolutionary dynamics of complex SV events under selection remains poorly established. Since many oncogenic alterations arise via complex SVs mentioned here, this is an important area. However, I have several major questions about the key assumptions, simulation settings, and validation.

First, in the model the DSBs are generated randomly throughout the genome in the G1 phase. In somatic cells, however, specific mechanisms such as fork collapse within the replication factory results in DNA breaks, which may violate the above assumption. There are more long range chromosomal interactions among the break ends than that expected by random chance. On a similar note, it may violate the assumption that pairing of DSBs is random, or merely linear distance-dependent along a chromosome. Ultimately, these assumptions can bias the context and size distribution of SVs expected under neutral evolution and selection (see PMID: 17137790; PMID: 21962511).

4.1 Response:

Because we focus on SVs generated by NHEJ, we primarily introduced DSBs in the G1 phase, where NHEJ predominates due to the absence of homologous sequences, such as sister chromatids, which are required for HR. Although replication-associated DSBs can occur in the S phase, such as those arising from fork collapse, they are more commonly repaired via HR (DOI: 10.1002/jcp.25048), such as microhomology-mediated break-induced repair (MMBIR). MMBIR is a common HR mechanism that was shown to repair stalled forks in haploinsufficient BRCA1 cells (DOI: 10.1016/j.molcel.2022.08.017) and to generate complex SVs and tandem duplications in cancer genomes (DOI: 10.1016/j.cell.2013.04.010). Recent studies show that cancers with chromothripsis have significantly fewer long homologies that are characteristic of HR, compared to cancers without chromothripsis (DOI: 10.1038/s41467-020-16134-7). Moreover, MMBIR contributes to a much smaller fraction of detected SVs than NHEJ (DOI: 10.1038/s41588-019-0576-7). Therefore, we assumed correct replication in the S phase to simplify the current model. We have included

incorporating replication-associated DSBs as future work in Discussion and clarified our choice of DSBs in G1 in more detail in Methods.

Although there are many long-range chromosomal interactions, loci on different chromosomes interact less frequently than those on the same chromosome, where interaction frequency generally decreases with increasing genomic distance, following a power-law decay (DOI: 10.1016/j.ymeth.2014.10.031). Analysis of Hi-C data from human cells showed that the lengths of chromosomal loops exhibit a power law distribution (DOI: 10.1126/science.118136). The observed lengths of focal CNAs also show a similar power law distribution (DOI: 10.1038/nbt.2049). Therefore, the reciprocal of the genomic distance between two breakpoints on the same chromosome provides a straightforward approximation of their spatial distance, eliminating the need for additional parameters such as the scaling factor for a power law distribution. We have included incorporating spatial distances for more accurate interaction frequencies of breakpoints as future work in Discussion.

Since the context and size distribution of SVs expected under neutral evolution are not well characterized, we analyzed the size distribution of SVs in real datasets, which have likely experienced selection pressures of varying, often unknown, directions and intensities. We simulated 500 datasets for each well-fit real dataset using parameters sampled from the posterior distributions of the inferred parameters and computed the size distributions of intrachromosomal SVs. We compared the median of these posterior distributions against the observed data for each real dataset using quantile-quantile (QQ) plots and Kolmogorov-Smirnov (KS) tests. Most simulated SV sizes were slightly larger than the observed values in single-cell datasets, among which two datasets have very similar simulated and observed distributions, as indicated by p-values from KS tests below 0.05 (Supplementary Fig. 22). For bulk datasets, the simulated SV sizes were mostly consistent or smaller than the observed values, among which 46 datasets have very similar simulated and observed distributions (Supplementary Fig. 28). Therefore, our model provides a realistic representation of SV sizes observed in real datasets.

Second, the model for chromothripsis assumes that DSBs are generated until n_d+1 cycles but none thereafter. It may be a convenient assumption to study evolutionary patterns of a chromothripsis event, but the assumption on subsequent genomic stability of the affected region may not be realistic. Rearranged genomic segments are likely to alter chromatin folding and replication timing contributing to persistent instability and heterogeneity at the affected locus. (example: Fig-8 of Voronina et al. Nature Comm, 2020).

4.2 Response:

We would like to clarify that our model can introduce DSBs in the G1 phase of every cycle when $n_d + 1$ equals or exceeds the maximum number of cell cycles, such as when n_d equals population size (Fig. 2f-g, Fig. 5-8). When $n_d + 1$ is smaller than the maximum number of cell cycles, subsequent genomic instability is modeled as simple breaks or local fragmentation during mitosis. Specifically, new DSBs can arise from the breakage of complex paths containing more than one centromere (Fig. 2b-e, 3-4). To investigate the mechanisms of chromothripsis, which often occurs over a short time frame, we allow the simulation until $n_d + 1$ cycles to focus on the consequences of cell division errors during the initial few cell cycles. This approach is intended to facilitate comparison with the findings in Umbreit et. al (DOI: 10.1126/science.aba0712), where a single cell division error generated genomic complexity and a cascade of mutations, including complex SVs. To avoid confusion, we have revised the text in Methods to explicitly state that DSBs will be introduced during G1 in every cell cycle when $n_d + 1$ equals or exceeds the maximum number of cell cycles.

We acknowledge that genome rearrangements can lead to subsequent genomic instability and affect several genomic properties, including chromatin folding and replication timing. Although these alterations in chromatin structure and replication timing may affect the occurrence and contact probabilities of DSBs, it is challenging to accurately measure these factors for estimating additional parameters. Therefore, our current model focuses solely on subsequent SVs resulting from chromosome mis-segregation. Given the practical difficulties of simultaneously obtaining Hi-C, replication timing, and DNA sequencing data from the same sample, we plan to incorporate DSB hotspots and contact probabilities derived from large-cohort public data in future work and have briefly added this point in Discussion.

Third, many oncogenes and tumor suppressor genes function in a tissue-dependent manner, such that the density of tumor suppressor and oncogenes to assess the clonal fitness may be simplistic. Chromosomal ploidy and local copy numbers may affect the pointy mutation rate in a cancer gene in the affected region, that has potential to ultimately changing the fitness landscape (e.g. copy number of EGFR and T790M mutation therein; also see Selmecki et al. Nature, 2015).

4.3 Response:

The OG-TSG score we used to measure cellular fitness already accounts for tissue-specificity as well as the effects of CNAs and point mutations. The OG-TSG score equals the negative Charm/Chrom (TSG-OG) score derived by Davoli et al. (DOI: [10.1016/j.cell.2013.10.011](https://doi.org/10.1016/j.cell.2013.10.011)). It is computed based on both the density and potency of tumour suppressor genes (TSGs) and oncogenic genes (OGs) on a specific chromosomal arm or chromosome. The potency of a gene is measured by its rank in respective lists obtained from the TUSON Explorer, which ranks genes according to their overall significance derived from mutational profiles. For OG prediction, two parameters were used, including 1) an entropy score which equals the weighted sum of probabilities that a site is mutated across a gene and 2) the ratio of high functional impact missense mutations to benign mutations. For TSG prediction, three parameters were used, including 1) the ratio of loss-of-function mutations to benign mutations, 2) the ratio of high functional impact missense mutations to benign mutations, and 3) the ratio of splicing mutations to benign mutations.

The Charm/Chrom (TSG-OG) score quantifies the positive or negative impact on growth and survival that wild-type OGs or TSGs typically confer to a chromosomal arm or whole chromosome. The score is higher for arms or chromosomes with a greater number or potency of TSGs and lower for those with a greater number or potency of OGs. A strong negative correlation exists between the score and amplification frequency. Therefore, this score reflects how CNAs influence cancer evolution by altering cumulative haploinsufficiency for deletions and cumulative triplosensitivity for amplifications.

The ploidy is not directly considered by the Charm/Chrom (TSG-OG) score, as the score focuses on the balance between OGs and TSGs. However, ploidy is implicitly integrated into the formula for the survival probability of each cell (Equation 1 in Methods). Ploidy affects the average total copy number of a chromosomal arm or chromosome in a cell, which in turn affects the fitness of the cell.

TUSON Explorer used pan-cancer data to identify driver genes and hence may overlook tissue-specific drivers. However, the analysis across 20 cancer types conducted by Davoli et al. suggests that most tissue-specific drivers were detected from pan-cancer data analysis, including approximately 70% of the TSGs detected in individual cancer types.

To clarify these points, we have revised the description of the OG-TSG score and included the reference to the original definition of the Charm/Chrom (TSG-OG) score in Methods. We also

consider improving the model of selection to explicitly incorporate tissue-specific context in future work and have added this point in Discussion.

Fourth, ecDNA copy number increase can confer fitness advantage, but at some point the high copy numbers result in fitness cost to the cell for maintaining the ecDNA (Verhaak et al. *Nature Rev Genet*, 2020, and references 51-52 therein). Likewise, unrepaired DSBs also result in a fitness cost to the cell. Such issues were not suitably modeled during simulation in this paper.

4.4 Response:

We agree that selection strengths in the model should vary accordingly to reflect the dynamics of ecDNA copy number changes. However, the exact ecDNA copy number at which selection pressure shifts direction represents a model parameter that is hard to estimate from available data. In Turner et. al., a threshold of 15 was used to indicate the decline in positive selection strength for ecDNAs carrying oncogenes, based on observations from cells with around 1000 extrachromosomal elements (DOI: 10.1038/nature21356). However, more recent studies suggest that ecDNA copy numbers show extreme cell-to-cell variations and depend on other external factors such as methotrexate treatment dose (DOI: 10.1038/s41588-022-01177-x). Unlike most existing models that focus solely on amplifications of ecDNAs carrying oncogenes, our model also includes ecDNAs carrying TSGs, which might be lost during cancer evolution due to negative selection. By using the OG-TSG score, our model captures both positive and negative effects of ecDNAs, based on their overlap with OGs and TSGs at their original genomic positions.

To demonstrate fitness changes of ecDNAs with high copy numbers in our model, we tracked one cell (ID: 197) with one ecDNA carrying cancer genes when the total number of cells reached 100 under selection (Extended Data Fig. 1b, Supplementary Fig. 9). This cell gave rise to two daughter cells (ID: 862 and 863), each carrying two ecDNAs overlapping cancer genes, which underwent negative and positive selection, respectively. Cell 862 had one ecDNA with a single copy and the other with two copies. Cell 863 had one ecDNA with four copies and the other with five copies. When the total number of cells reached 5000, cell 862 had eight descendants, all of which but one underwent negative selection. In contrast, cell 863 had 13 descendants, all of which underwent positive selection. As indicated by node sizes, cells with higher ecDNA copy numbers generally had a higher probability of survival, but they also sometimes underwent weaker positive selection (e.g., cell 6227 with copy number 10 for ecDNAs overlapping cancer genes) or negative selection (e.g., cell 9479 with copy number 4 for ecDNAs overlapping cancer genes).

We also acknowledge that incorporating the fitness cost of unrepaired DSBs is essential for developing a more realistic model. The percentage of unrepaired DSBs is a key parameter in our model. Unrepaired DSBs mainly lead to telomere fusions and BFB cycles, which in turn generate CNAs and SVs. As it is challenging to directly measure unrepaired DSBs and their cytotoxicity from real cancer data, we use CNAs and SVs detected from whole-genome sequencing data as indirect proxies to approximate the fitness of unrepaired DSBs.

The OG-TSG score we used to measure fitness takes into account CNAs and point mutations within. We did not explicitly model the fitness of balanced SVs other than amplifications and deletions. Balanced SVs may cause gene fusions or disrupt regulatory elements, which often require additional data, such as gene expression, to measure fitness (DOI: 10.1038/s41568-022-00488-9). As we focus on estimating DSB-related parameters from CNAs and SVs detected in cancer genomes in this study, we used a simplified selection model with only one parameter for selection strength.

To develop more realistic selection models, we will incorporate additional factors affecting selection as parameters in future work, such as maximum ecDNA copy number advantageous for cell survival and fitness cost of unrepaired DSBs or balanced SVs, as richer data become available. We have included these considerations in Discussion.

Fifth, validation analyses could be more rigorous, ideally using data from barcoded cell populations from in vitro experiments, or deeply profiled human tumors (with high mutation burden) to secure deep clonal phylogeny. Fig-6 presented a valuable analysis, but I felt that the pseudobulk analysis, as presented, under-utilized the potential of the dataset. However, I was not convinced by the interpretation that dismissed unfavorable results as - 'For a few datasets, the inferred DSB rates were more different from empirical DSB rates (Extended Data Fig. 7a), which could be due to SVs generated from mechanisms not captured in our model such as HRD or replication errors'. Again, these issues call for more validation analyses from different tumor types.

4.5 Response:

As our inferences use summary statistics derived from detected breakpoints and SVs, pseudobulk analysis is necessary to obtain breakpoints at nucleotide resolution from low-coverage single-cell sequencing data. In the original pseudobulk approach, a breakpoint is considered present in a clone if at least one cell in the clone supports it, which provides sufficient power to recover breakpoints at a cumulative coverage of 5x (DOI: 10.1038/s41586-022-05249-0). We have rephrased the justifications for using pseudobulk data in the main manuscript to avoid confusion.

To better illustrate the uncertainties in Bayesian inferences, we replotted Extended Data Fig. 7 using 95% credible intervals instead of standard deviations. We define a dataset as poorly fit when its empirical DSB rate falls outside the 95% credible interval of the inferred DSB rate by a considerable margin (i.e., absolute differences to the interval boundaries larger than 10). One dataset, DG1134, was identified as poorly fit, with an acceptance rate nearly ten times higher than the other datasets, all of which reached the target tolerance of 0.5. To improve the accuracy of the inferences, we reduced the target tolerance to 0.2 and reran ABC SMC for DG1134. Despite this adjustment, the result still showed a poor fit, likely due to the presence of approximately 27% tandem duplications. To ensure more reliable results, we excluded DG1134 from the final analysis (Fig. 6). We also replaced the ambiguous reasoning with the exact fractions of tandem duplications in the main manuscript.

Deeply profiled human tumours with single-cell sequencing and clonal phylogeny data are rare. Therefore, we used deeply profiled bulk sequencing datasets from the Pan-Cancer Analysis of Whole Genomes (PCAWG) Consortium. These datasets have mean read coverages that follow a bimodal distribution with modes at 38x and 60x. PCAWG datasets have also been extensively studied, revealing multiple types of complex SVs including chromothripsis, BFBs, and ecDNAs. We enhanced our simulation program to allow inferences on bulk whole-genome sequencing data and applied Bayesian inferences to these datasets. Among the 111 samples analyzed using ABC SMC, 82 samples exhibited a good fit, whereas 29 samples were poorly fit, likely due to significantly more tandem duplications. We have added the results in a new section titled "Fitting the model to bulk whole-genome sequencing data of multiple cancer types" in the main manuscript (Fig. 7-8).

REVIEWER COMMENTS

Reviewer #2 (Remarks to the Author):

In general, very happy with authors' response. Some minor issues remain:

In both the responses to the reviews and the paper, it has been repeatedly mentioned that time constraints have affected the ability to optimize the tolerance parameter, thereby impacting result accuracy. Please provide an estimated runtime for the inference process to give a clearer understanding of the time requirements.

Response:

We set a 24h time limit for inferences on simulated data and a 72h time limit (the maximum wallclock on the HPC cluster) for inferences on real data. We have now added the runtime limit in section "Running ABC SMC on simulated data and whole-genome sequencing data".

In Figure 7, the subfigures are misaligned and lack proper labelling. Additionally, the colours used are not easily distinguishable, making it difficult to interpret the data effectively.

Response:

We have now aligned the subplots and adjusted the colors for better distinction in Figure 7. To reduce visual clutter, we only retained x-axis labels for the bottom subplot and removed the bolding which was used to indicate samples with CCNE1 amplification. Additionally, we added the subplot for the inferred probability of WGD in the middle, which makes it easier to identify samples with WGD across the five subplots. As the subplots are better aligned, it should be straightforward to locate the sample name corresponding to each violin plot.

Figure 7: How are you calculating the number of ecDNA per cell in the bulk data? What assumptions or estimates did you use to determine the number of cells?

Response:

The number of ecDNAs per cell in the bulk data is inferred based on simulation settings where individual cells were simulated and treated as clones. As in Figure 6, parameter interpretations were adjusted to reflect clone expansion rather than cell division. To maintain consistency, we have retained the same labels. Specifically, the reported number of ecDNAs per cell actually represents the inferred number of ecDNAs per clone. To clarify this distinction, we have now explicitly restated it in section "Fitting the model to bulk whole-genome sequencing data of multiple cancer types".

Figure 8:

report the correlation and p-value for all the subplots.

Response:

We did not previously show correlation coefficients and corresponding p-values for subplots **a**, **b**, **e**, and **f** because the x- and y-axis variables measure very similar quantities and are inherently expected to be correlated. In subplots **a** and **b**, the linear regression lines illustrate the linear relationships, so reporting Spearman correlation coefficients, which capture monotonic trends, would not provide additional meaningful insights. In subplots **e** and **f**,

since no clear correlation is observed, the correlation coefficients and p-values are expected to be nonsignificant.

For completeness and consistency, we have now included Spearman correlation coefficients and corresponding p-values for subplots **a**, **b**, **e**, and **f**. To be more precise, we have also replaced “correlation” in the legend to “relationship”.

What criteria are used to define well-fit samples in both single-cell and bulk datasets?

Response:

Well-fit samples are those not poorly fit, namely when its empirical DSB rate falls within or outside the 95% credible interval of the inferred DSB rate by a small margin, with absolute differences to the interval boundaries no larger than 10. We have now added its definition at the end of Methods to avoid confusion.

Comparing the number of ecDNA in single-cell and bulk datasets in section “Fitting the model to bulk whole-genome sequencing data of multiple cancer types” does not seem feasible. This is because the number of ecDNA elements is highly dependent on the type of sample being analysed. It is essential to specify the types of samples used in Figure 8 for both single-cell and bulk datasets. For example, a higher percentage of ecDNA is expected in late-stage oesophageal adenocarcinoma compared to early-stage cases (DOI: 10.1038/s41586-023-05937-5).

Response:

It would be more meaningful to compare the inferred number of ecDNAs in a dataset against the actual or expected number. However, it remains challenging to accurately detect ecDNAs from whole-genome sequencing data, particularly single-cell data.

As ecDNAs in database eccDNAdb were obtained from tumour samples, including patient tissues, PDXs, and cancer cell lines, the number of overlapping ecDNAs from eccDNAdb provides an upper bound on the potential number of ecDNAs for the same cancer type.

To facilitate better interpretation of the results, we have now assigned colors to data points based on sample type in Figure 8. Specifically, we colored the 20 single-cell datasets by cell type in subplot **a** and the 82 bulk datasets by sample type in subplot **b-f**. To improve visualization of overlapping data points, we applied jittering with both height and width set to 0.1. We also added sample types in Supplementary Table 8.

With simulated data, it is shown that the number of ecDNAs per cell is significantly higher in cells with WGD compared to those without WGD. Similarly, in the real data, the section “Fitting the model to bulk whole-genome sequencing data of multiple cancer types” suggests that the posterior mean number of ecDNA significantly increases with the inferred probability of WGD. However, in Figure 7, the samples with WGD appear to have a lower inferred mean number of ecDNAs. Could you clarify this?

Response:

We apologize for a mistake in the figure legend, which should state “datasets without WGD are shown in grey”. Since most datasets with WGD have violin plots positioned at the top of the subplot, it is evident that samples with WGD tend to have a higher inferred mean number of ecDNAs than those without WGD.

To reduce crowding and improve plot alignment without additional x-axis labels, we previously removed the subplot for inferred probability of WGD. However, this made it less clear which datasets have WGD without reading the labels. We have now corrected the legend, added the WGD subplot, and clarified that 29 well-fit datasets had WGD by providing additional details on the excluded datasets in section “Fitting the model to bulk whole-genome sequencing data of multiple cancer types”.

Please clearly define the specific patterns you considered as chromothripsis-like patterns and elaborate on what you mean by chromothripsis-related events in the results section to enhance clarity.

Response:

We used “chromothripsis-like patterns” to describe the patterns detected by ShatterSeek (Methods), which are referred to as chromothripsis (events) elsewhere in the manuscript. We intended to distinguish between patterns of chromothripsis detected by ShatterSeek and real chromothripsis events. To maintain consistency in terminology, we have now revised “chromothripsis-like patterns” to “chromothripsis”.

We used “chromothripsis-related events” to describe events likely to lead to chromothripsis, such as local fragmentation, which can trigger complex chromosomal rearrangements including chromothripsis (DOI: 10.1126/science.aba0712). To improve clarity, we have revised the text to be more specific and replaced “chromothripsis-related” with “local fragmentation”, followed by explanations.

Reviewer #3 (Remarks to the Author):
